# LiFR-Seg: Anytime High-Frame-Rate Segmentation via Event-Guided Propagation

**Xiaoshan Wu, Xiaoyang Lyu**[*]**, Yifei Yu & Bo Wang**
Department of Electrical and Computer Engineering
The University of Hong Kong
`{xiaoshan,shawlyu,yfyu,bowang98}@connect.hku.hk`

**Zhongrui Wang**[†]
School of Microelectronics
Southern University of Science and Technology
`wangzr@sustech.edu.cn`

**Xiaojuan Qi**[†]
Department of Electrical and Computer Engineering
The University of Hong Kong
`xjqi@eee.hku.hk`

## Abstract

Dense semantic segmentation in dynamic environments is fundamentally limited by the low-frame-rate (LFR) nature of standard cameras, which creates critical *perceptual gaps* between frames. To solve this, we introduce *Anytime Interframe Semantic Segmentation*: a new task for predicting segmentation at any arbitrary time using only a single past RGB frame and a stream of asynchronous event data. This task presents a core challenge: how to robustly propagate dense semantic features using a motion field derived from sparse and often noisy event data, all while mitigating feature degradation in highly dynamic scenes. We propose LiFR-Seg, a novel framework that directly addresses these challenges by propagating deep semantic features through time. The core of our method is an *uncertainty-aware warping process*, guided by an event-driven motion field and its learned, explicit confidence. A *temporal memory attention* module further ensures coherence in dynamic scenarios. We validate our method on the DSEC dataset and a new high-frequency synthetic benchmark (SHF-DSEC) we contribute. Remarkably, our LFR system achieves performance (73.82% mIoU on DSEC) that is statistically indistinguishable from an HFR upper-bound (within 0.09%) that has full access to the target frame. We further demonstrate superior robustness across extreme scenarios: in highly dynamic (M3ED) tests, our method closely matches the HFR baseline's performance, while in the low-light (DSEC-Night) evaluation, it even surpasses it. This work presents a new, efficient paradigm for achieving robust, high-frame-rate perception with low-frame-rate hardware.

## 1 Introduction

Accurate and dense semantic understanding of dynamic scenes is a critical capability for autonomous systems, including self-driving cars, drones, and robotics Dovesi et al. (2020); Tsai et al. (2023). The prevalent paradigm, however, relies on conventional RGB cameras that capture information at discrete, often low, frequencies (e.g., 20 Hz). This low sampling rate creates significant "*perceptual gaps*", or "*blind spots*", between frames. This results in a crucial "blind time interval" during which fast-moving or abruptly appearing objects are not perceived, posing a severe risk in high-speed scenarios Gallego et al. (2020); Guo et al. (2018).

The scenario in Fig. 1 illustrates this danger: a pedestrian suddenly enters a vehicle's path. At timestamp $t$, the Low-Frame-Rate (LFR) system (Fig. 1b) detects no hazard. In the very next frame, at timestamp $t + \Delta t$, the pedestrian appears, but it is already too late for the autonomous system to react. This motivates the need for a High-Frame-Rate (HFR) perception system that can produce

---

[*]Project lead.
[†]Corresponding author.

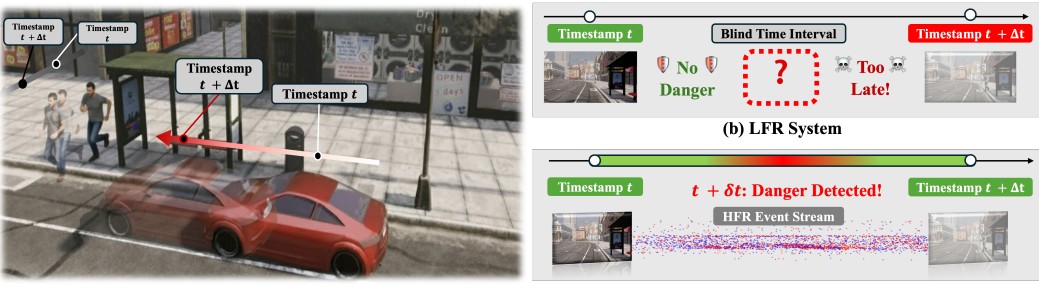

Figure 1: **Bridging the Perceptual Gap in High-Speed Scenarios.** A critical "Blind Time Interval" for LFR systems is illustrated: **(a)** During $t$ to $t + \Delta t$, a pedestrian rapidly enters the vehicle's path. **(b)** A standard LFR system, constrained by discrete frames, *Too Late* detects danger only by $t + \Delta t$. **(c)** In stark contrast, our HFR Anytime System leverages continuous events to detect imminent danger at $t + \delta t$, providing crucial early warning and bridging this gap.

reliable predictions **at any moment in time**. To formalize this ambition, we propose a new task, **Anytime Interframe Semantic Segmentation**. While high-speed RGB cameras could mitigate this gap, their prohibitive cost, high power consumption, and massive data bandwidth requirements make them impractical for scalable, real-world deployment (detailed in Appendix D).

Event cameras, in contrast, record brightness changes asynchronously at microsecond resolution Gallego et al. (2020); Berlincioni et al. (2023); Wang et al. (2024), naturally capturing high-temporal-resolution motion while consuming far less power and bandwidth. However, an event camera is a temporally dense but spatially sparse camera, which limits its semantic information. Thus RGB camera cooperating with event streams offers a more practical and scalable way to build an HFR perception system.

Formally, we define the anytime interframe semantic segmentation task as predicting a dense semantic map at *any arbitrary timestamp* $t + \delta t$ within a perceptual gap $(t, t + \Delta t]$, given only the initial RGB frame $I_t$ and the corresponding event stream $\mathcal{E}_{t-\Delta t \rightarrow t+\delta t}$. This formulation imposes two critical constraints that differentiate our work from standard paradigms: **Causality** (requiring no future frames like $I_{t+\Delta t}$) and **Anytime Prediction** (predicting for *any* $\delta t$, not just at fixed frame times). As we will illustrate in our baseline comparison (Fig. 3), existing paradigms fail one or both of these constraints: standard video interpolation is often *non-causal*, while multi-modal fusion methods are typically *not anytime-capable*. Our work presents the first framework designed to satisfy both.

This task presents a non-trivial challenge: how to effectively merge the rich, static semantic context from the past RGB frame $I_t$ with the temporally dense, but spatially sparse and often noisy, event stream $E_{t-\Delta t \rightarrow t+\delta t}$? Our core insight is to leverage the continuous event stream to estimate a **high-frequency motion field** (§3.2), which serves as a robust bridge to temporally propagate the deep semantic features from $I_t$ to the target time $t + \delta t$. To achieve this, our framework, **LiFR-Seg**, introduces three key technical designs. First, we operate on and propagate multi-scale **deep semantic features**, as this preserves semantic detail. Second, we integrate an **uncertainty-aware warping mechanism** (detailed in §3.3) that explicitly learns to modulate the feature propagation based on the estimated motion reliability. Finally, to ensure temporal consistency and handle occlusions over long prediction intervals, we incorporate a **temporal memory attention** module (§3.4).

We rigorously evaluate our framework on a comprehensive benchmark comprising the real-world DSEC dataset Gehrig et al. (2021a) and a new, high-frequency synthetic dataset (SHF-DSEC) that we contribute. Our experiments show that combining a low-frame-rate RGB camera with asynchronous event streams achieves performance on par with a fully high-frame-rate system. On DSEC, our method, despite having *no access* to the interframe RGB data at $t + \delta t$, still achieves 73.82% mIoU, demonstrating a **gap of less than 0.09%** compared to the 73.91% mIoU achieved by an ideal **HFR upper-bound model** (§5.1) with full access to the target frame. Furthermore, we demonstrate strong robustness in challenging scenarios: on the highly dynamic M3ED dataset, our approach outperforms all baselines, and in zero-shot evaluation on the DSEC-Night benchmark, it **even surpasses the upper bound**, confirming its efficacy in high-motion and low-light conditions.

Our contributions are threefold:

- We introduce **Anytime Interframe Semantic Segmentation**, a novel and practical task for perception in dynamic environments, effectively bridging the "perceptual gap" inherent in standard camera systems.

- We propose a multi-modal framework that robustly propagates semantics from a single RGB frame using event-driven motion cues. Key components include uncertainty-aware feature warping and a temporal memory mechanism.

- We release a new high-frequency synthetic dataset (SHF-DSEC) and establish a strong benchmark. We demonstrate state-of-the-art performance, showing that our low-frame-rate system matches a high-frame-rate upper bound and excels in high-dynamic and low-light scenarios.

## 2 RELATED WORK

**Video Semantic Segmentation (VSS)** Video Semantic Segmentation (VSS) leverages temporal coherence to improve segmentation consistency and efficiency across frames Guo et al. (2018); Mo et al. (2022). Early methods focused on spatial feature extraction Yu et al. (2018); Li et al. (2022), while more recent approaches explicitly propagate information. For instance, Deep Feature Flow Zhu et al. (2017) uses optical flow to warp features from keyframes to subsequent frames, but this is primarily for *acceleration* of an already dense video stream. Other modern methods Ravi et al. (2024) integrate temporal memory to exploit motion cues. However, all these methods are fundamentally **frame-based**. They presuppose a dense, high-frame-rate (HFR) RGB video stream as their input. They are not designed to solve the "perceptual gap" problem we address: predicting segmentation at an arbitrary time $t + \delta t$ using only a *single* past RGB frame from time $t$.

**Event-based Vision and Segmentation** Event cameras offer an alternative sensing modality, capturing pixel-level brightness changes asynchronously with high temporal resolution and high dynamic range Gallego et al. (2020); Berlincioni et al. (2023). This makes them ideal for capturing motion. However, their data is spatially sparse and lacks the rich semantic texture of RGB frames. Consequently, methods for **event-only** semantic segmentation Alonso & Murillo (2019); Binas et al. (2017); Zhu et al. (2021) often struggle to produce dense, high-fidelity semantic maps, which highlights the clear necessity of a multi-modal approach that includes RGB context.

**Multi-Modal RGB-Event Fusion and Propagation** To leverage the strengths of both sensors, several multi-modal frameworks have been proposed. One dominant paradigm is **feature fusion**. Methods like CMNeXt Zhang et al. (2023) and EISNet Xie et al. (2024) (which we compare against in Sec. 5) use parallel encoders to extract features from both RGB and event data, then fuse them using attention or concatenation. However, these methods are designed for *enhancement*—that is, using events from $t - \Delta t$ to improve the segmentation of the RGB frame *at* time $t$. They do not perform **temporal propagation** into a future "blind gap" from a single $I_t$, which is the core of our task.

A more relevant paradigm is **temporal propagation** using event-based optical flow Wan et al. (2022); Gehrig et al. (2024). The concept of feature warping via flow is established in VSS Zhu et al. (2017) and video interpolation Niklaus & Liu (2020). However, these prior works typically warp information between two *known* RGB frames, requiring a future frame $I_{t+\Delta t}$. Our work is the first to leverage the unique properties of event-based flow to propagate **deep semantic features** from a *single* past RGB frame to an *arbitrary future timestamp*. We further innovate by introducing an uncertainty-aware mechanism to handle flow inaccuracies from sparse events and a memory module to ensure long-term temporal consistency.

## 3 METHOD

### 3.1 FRAMEWORK OVERVIEW

Our goal is to solve the task of **Anytime Interframe Semantic Segmentation**. We formally define this as estimating the dense semantic label probability distribution $P(\text{Seg}_{t+\delta t}|I_t, \mathcal{E}_{t-\Delta t \to t+\delta t})$ for a

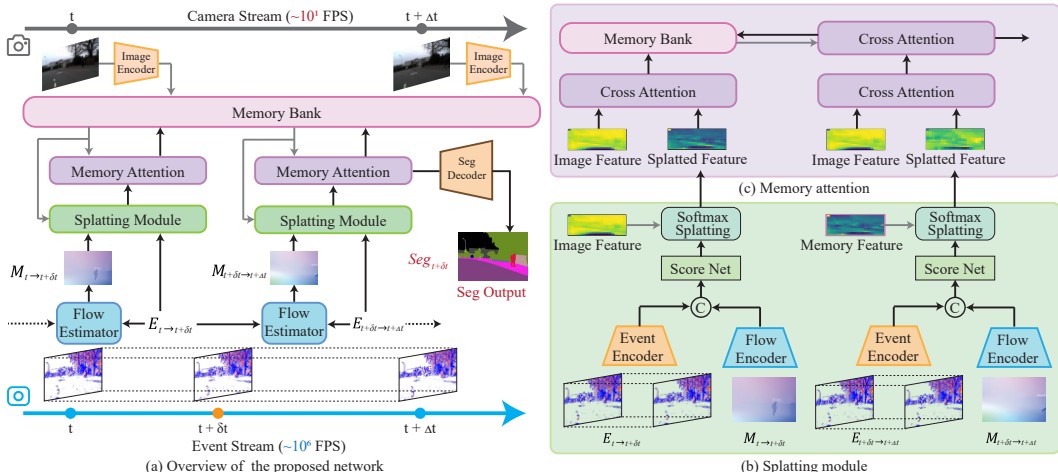

Figure 2: **Overview of our LiFR-Seg framework. (a)** The overall architecture. **(b)** The **Splatting Module** performs uncertainty-guided feature propagation using an event-driven motion field ($\hat{\mathbf{M}}$) and its learned confidence ($S$). (Note that $E_{t+\Delta t}$ is used strictly for training supervision to generate $Seg_{t+\Delta t}$.) **(c)** The **Memory Attention** module refines the propagated feature by integrating historical context for long-term consistency.

target time $t + \delta t$. We define $\Delta t$ as the fixed interval between consecutive LFR frames (e.g., 50ms), and $\delta t \in (0, \Delta t]$ as the relative offset to the target timestamp within this interval. This is a challenging *spatio-temporal prediction problem*: we must infer the dense, per-pixel semantic state ($Seg_{t+\delta t}$) conditioned only on a single, **spatially dense but temporally sparse** image observation ($I_t$) and the **temporally dense but spatially sparse** stream of intermediate motion cues ($\mathcal{E}_{t-\Delta t \to t+\delta t}$).

As illustrated in Fig. 2, our framework decomposes this problem into three core stages. First, to robustly model the scene dynamics, we estimate an **Event-Driven Uncertainty-Aware Motion Field** (§3.2). Second, using this motion field, we perform **Uncertainty-Guided Feature Propagation** to warp the initial deep features to the target timestamp (§3.3). Finally, to ensure **Long-Term Consistency**, the propagated features are refined using a temporal memory module (§3.4).

## 3.2 EVENT-DRIVEN UNCERTAINTY-AWARE MOTION FIELD

To propagate dense features, we require a dense motion field. We estimate this field from the raw, asynchronous event stream, denoted as $\mathcal{E}$. However, this estimation from sparse data has **inherent uncertainty**. Therefore, we model the true (but unknown) motion field $\mathbf{M}$ probabilistically. Our goal is to estimate not only its mean (the flow vector $\hat{\mathbf{M}}$) but also its precision (a confidence score $S$).

First, the raw, asynchronous event stream $\mathcal{E}$ within a specific time window is converted into a discrete, grid-based representation, the event voxel $E$. This is achieved by accumulating the polarity of events into several temporal bins. The value for a given pixel $\mathbf{u} = (x, y)$ and bin index $b$ is computed as:

$$E(\mathbf{u}, b) = \sum_{e_j \in \mathcal{E}} p_j \cdot [\mathbf{u}_j = \mathbf{u}] \cdot \max(0, 1 - |t_j^* - b|), \tag{1}$$

where $[\cdot]$ denotes the *Iverson bracket*, which is 1 if the condition inside is true and 0 otherwise. $p_j$ is the event polarity (+1 or -1). The term $t_j^* = \frac{(B-1)(t_j - t_0)}{\Delta \mathcal{T}}$ is the normalized event timestamp, where $B = 4$ is the total number of bins, $b \in \{0, \ldots, B-1\}$ is the specific bin index, $\Delta \mathcal{T}$ is the time window, and $t_0$ is its start time. These voxels, $E_{t-\Delta t \to t}, E_{t \to t+\delta t} \in \mathbb{R}^{B \times H \times W}$, are then fed into an event-based optical flow network to predict the conditional mean of the motion field, $\hat{\mathbf{M}}_{t \to t+\delta t} \in \mathbb{R}^{2 \times H \times W}$. This network, denoted as $\mathcal{F}_{FlowNet}$, follows a modern RAFT-like architecture Teed & Deng (2020). First, a feature encoder ($\phi_{feat}$) is applied once to extract features from each event voxel, which are used to build a 4D correlation volume $\mathcal{V}_{corr}$. Then, starting from an initial estimate $\hat{\mathbf{M}}^0 = \mathbf{0}$, an update operator ($\mathcal{U}_{update}$) iteratively refines the flow for $k = 0, \ldots, K-1$ steps:

$$\hat{\mathbf{M}}^{k+1} = \mathcal{U}_{update}(\hat{\mathbf{M}}^k, \mathcal{C}(\hat{\mathbf{M}}^k, \mathcal{V}_{corr})). \tag{2}$$

where $\mathcal{C}$ is the correlation lookup operator. The final $\hat{\mathbf{M}}_{t \to t+\delta t}$ is the output of the last iteration, $\hat{\mathbf{M}}^K$.

Next, to estimate the reliability of this prediction, we introduce a **ScoreNet** (Fig. 2b) that learns a confidence map $S$, which serves as the log-precision of the flow distribution. The ScoreNet function, $\mathcal{F}_{ScoreNet}$, maps the input pair $(E_{t \to t+\delta t}, \hat{\mathbf{M}}_{t \to t+\delta t})$ to a single-channel log-precision map $S \in \mathbb{R}^{1 \times H \times W}$. This function is a composition of three main stages. First, separate encoders extract features from the event voxel and motion field: $F_E = \phi_{\text{event}}(E_{t \to t+\delta t})$ and $F_M = \phi_{\text{flow}}(\hat{\mathbf{M}}_{t \to t+\delta t})$. These are fused into a joint embedding $F_{\text{joint}} = \text{Concat}(F_E, F_M)$. Finally, the ScoreNet processes this embedding to regress the pixel-wise log-precision map $S$:

$$S_{t \to t+\delta t} = \psi_{\text{ScoreNet}}(F_{\text{joint}}) \tag{3}$$

This log-precision map $S$ is critical, as it serves as a key input to our feature propagation module (§3.3), where it will modulate the influence of each flow vector in a manner analogous to a weighted likelihood estimation.

### 3.3 Uncertainty-Guided Feature Propagation

With the estimated motion field $\hat{\mathbf{M}}$ and its confidence map $S$, we can now address the core task of temporally propagating semantic features. Our core design choice is to warp the multi-scale features, $F_t$, extracted from the LFR RGB-based segmentation backbone. As confirmed by our ablation study (Table 4), this strategy is superior to warping raw images or final segmentation maps.

This propagation is performed using Softmax Splatting Niklaus & Liu (2020). We make this operation **uncertainty-guided** by incorporating our event-guided confidence map $S$ as the log-space importance weight:

$$F_{t+\delta t} = \frac{\overrightarrow{\Sigma}(\exp(S_{t \to t+\delta t}) \cdot F_t, \hat{\mathbf{M}}_{t \to t+\delta t})}{\overrightarrow{\Sigma}(\exp(S_{t \to t+\delta t}), \hat{\mathbf{M}}_{t \to t+\delta t})}. \tag{4}$$

This ensures that features warped by unreliable flow vectors (low $S$) are given less "vote" in the final propagated feature map. To further correct for any residual warping artifacts, we then apply a lightweight **RefineNet**, composed of two sequential convolutional layers, which acts as a learned spatial regularizer to enhance the final feature consistency before decoding.

### 3.4 Long-Term Consistency via Temporal Memory

The feature propagation described so far is a **Markovian process**—the state at $t + \delta t$ depends only on the state at $t$. This is insufficient for real-world scenarios involving long temporal gaps or complex occlusions, where long-term context is required. To overcome this limitation, we introduce a memory mechanism to integrate this non-Markovian history.

We model this as a **recurrent state update**. A memory bank $\mathcal{M}$ stores features from previous key timestamps. As shown in Fig.2c, after a feature $F_{t+\delta t}^{deep}$ is generated via warping, it undergoes a **temporal enhancement** step. The propagated feature queries the entire memory bank $\mathcal{M}$ via cross-attention, producing an updated feature that is enriched with available historical context. This updated feature is then stored back in $\mathcal{M}$ for future use. We apply this mechanism only to the deepest, most semantic feature layer ($F^{\text{deep}}$) to effectively balance performance and computational cost, a choice validated by our experiments (Table 5) which show its critical role in long-interval robustness.

## 4 Benchmark

We evaluate our framework on four diverse datasets to assess its performance across a range of real-world conditions. For training and primary evaluation, we use the real-world DSEC dataset Gehrig et al. (2021a) along with our newly introduced SHF-DSEC, both of which focus on autonomous driving scenarios. **Specifically, SHF-DSEC features a higher temporal resolution of 100 Hz, enabling effective demonstration of our method's anytime prediction capability.** To further validate the robustness of our method across different domains, we include the M3ED dataset Chaney et al. (2023), which features sequences captured from drones and quadruped robots. Additionally, we test on the DSEC-Night benchmark Xia et al. (2023) to demonstrate the effectiveness of our approach under extreme low-light conditions. Please check the Appendix A for detailed information.

## 5 EXPERIMENTS

### 5.1 MAIN RESULTS

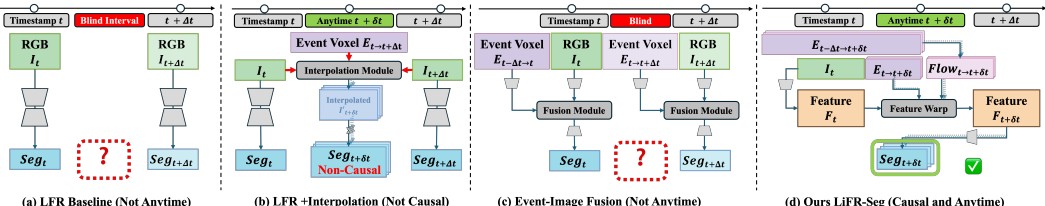

Figure 3: **Perception Paradigm Comparison.** Visual definition of the four experimental settings: **(a)** The **LFR (Baseline)**, which is causal but not anytime-capable. **(b) Interpolation-based** methods, which are **non-causal**. **(c)** The original **Event-Image Fusion** paradigm. **(d)** Our (LiFR-Seg) framework, which is the only one that is both **causal** and **anytime-capable**.

**Baseline Paradigm Comparison.** We establish our experimental setting by defining the key baseline paradigms, which are visually compared in Fig. 3 and quantitatively evaluated in Table 1. We first define two theoretical bounds: the HFR Upper Bound, representing an ideal (but impractical) system with access to the privileged target frame $I_{t+\delta t}$, and the **LFR (Baseline)** (Fig. 3a), a naive causal method using only $I_t$, which is not anytime-capable and thus suffers from a "**Perceptual Gap**". We then evaluate two competing paradigms. **Interpolation-based** methods (Fig. 3b) are inherently **non-causal**, as their core design requires the future frame $I_{t+\Delta t}$, making them incompatible with the causal constraints of our predictive task. The standard **Event-Image Fusion** paradigm (Fig. 3c) (e.g., CMNeXt Zhang et al. (2023)) is causal but not inherently *anytime-capable*; it is designed to fuse $I_t$ with co-located (often past) events (e.g., $E_{t-\Delta t \to t}$) to enhance $Seg_t$. To create a robust **LFR + Fusion** baseline for our task, we **adapted this paradigm** by providing it with $I_t$ and the *forward-looking* event stream $E_{t \to t+\delta t}$. This adapted approach, however, still suffers from the fundamental **limitation of direct fusion**: it struggles to effectively merge spatially dense semantic features (from RGB) with sparse, low-texture motion cues (from events), which can degrade segmentation accuracy. In stark contrast, our **LiFR-Seg** framework (Fig. 3d) is designed from the ground up to be both fully **causal** and truly **anytime-capable**, robustly *propagating* features (rather than fusing them) by leveraging the *full* available event context ($I_t$ and $E_{t-\Delta t \to t+\delta t}$).

**Experiment Setup.** To rigorously validate our framework, we conducted extensive evaluations on the real-world DSEC and M3ED datasets, our synthetic SHF-DSEC dataset, and the DSEC-Night benchmark. For a fair comparison, all methods leverage the same **Segformer-B2** backbone and are trained to convergence on their respective datasets, with the exception of the zero-shot DSEC-Night evaluation. We employ the OhemCrossEntropy loss Shrivastava et al. (2016) for end-to-end training to handle class imbalance. Supervision is applied at timestamps $t + \Delta t$, aligned via the second warping strategy (details in Appendix C.1).

**Quantitative Comparison.** The quantitative results, presented in Table 1, confirm the limitations of these baseline paradigms and reveal how our method successfully bridges the perceptual gap. On the standard **DSEC** dataset, our approach's efficacy is highlighted by its proximity to ideal performance; at **73.82%** mIoU, it closes the performance gap to the HFR Upper Bound (73.91%) to a mere **0.09%**, despite having no access to the target RGB frame. This trend of near-HFR performance continues on SHF-DSEC and the M3ED-Quadruped dataset. The advantage of our propagation mechanism is particularly pronounced in high-speed scenarios; on **M3ED-Drone**, our method attains **64.28%** mIoU, a remarkable improvement of **9.05%** over the LFR (Baseline) (55.23%). Most strikingly, our framework demonstrates unparalleled robustness in extreme low-light conditions. In the zero-shot **DSEC-Night** test, our approach (**41.86%**) not only functions effectively where the RGB-only HFR Upper Bound collapses (41.83%) but even **surpasses it**. This pivotal result proves that our event-driven system is not just a substitute for, but can be superior to, HFR-RGB systems when traditional vision fails.

Next, we evaluate against the **Interpolation-based** paradigm (e.g., TLX + Seg.), which is constrained by two fundamental limitations. Architecturally, it is **non-causal**, as its core design requires the future frame $I_{t+\Delta t}$ (Table 1), precluding its use in predictive scenarios. Performance-wise, it suffers from a mismatch between photometric reconstruction and semantic understanding. These flaws are

Table 1: Comprehensive performance comparison across five diverse benchmarks. Our method is the only one satisfying the crucial *causal* (CS) and *anytime* (AT) constraints for real-world prediction. It not only bridges the perceptual gap by matching HFR performance on standard datasets but also demonstrates superior robustness in high-speed (M3ED) and low-light (DSEC-Night) scenarios. Results are reported at $\delta t = 50$ms (DSEC, SHF, Night) and $\delta t = 40$ms (M3ED).

| Method | Input | CS | AT | DSEC | SHF | M3ED-D | M3ED-Q | D-Night |
|---|---|---|---|---|---|---|---|---|
| **HFR (Ideal)** | $I_{t+\delta t}$ | ✗ | ✗ | 73.91 | 65.40 | 64.57 | 69.27 | 41.83 |
| **LFR (Baseline)** | $I_t$ | ✓ | ✗ | 67.67 | 61.73 | 55.23 | 63.20 | 37.44 |
| **LFR + Interpolation** | | | | | | | | |
| TLX + Seg. | $I_t, \mathbf{I_{t+\Delta t}}, E...$ | ✗ | ✓ | 68.17 | 55.89 | 60.60 | 62.92 | NaN |
| **LFR + Fusion** | | | | | | | | |
| EISNet | $I_t, E_{t\to t+\delta t}$ | ✓ | ✓ | 68.11 | 61.28 | 58.34 | 62.98 | 37.28 |
| CMNeXt | $I_t, E_{t\to t+\delta t}$ | ✓ | ✓ | 70.13 | 61.40 | 59.56 | 65.52 | 39.38 |
| **Ours** | $I_t, E_{t-\Delta t\to t+\delta t}$ | ✓ | ✓ | **73.82** | **64.80** | **64.28** | **68.89** | **41.86** |

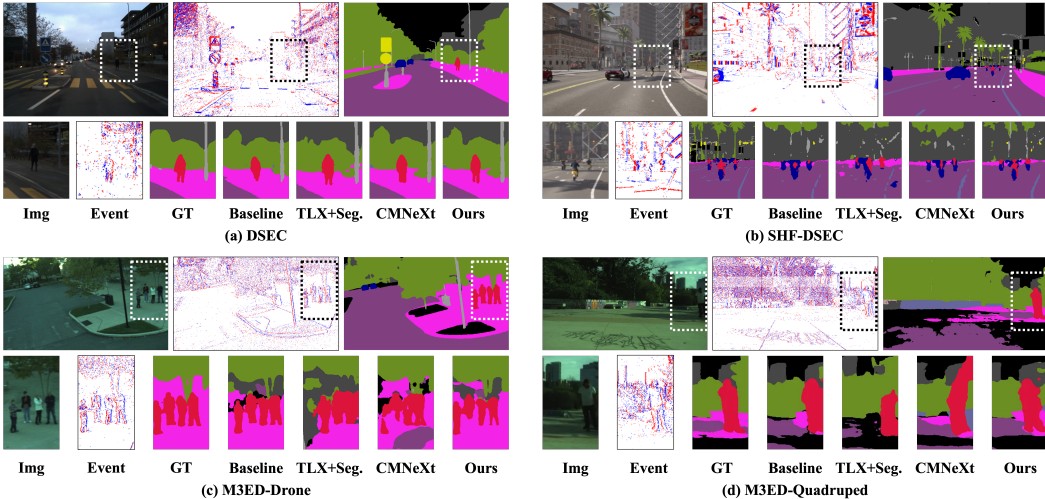

**(a) DSEC**     **(b) SHF-DSEC**     **(c) M3ED-Drone**     **(d) M3ED-Quadruped**

Figure 4: **Qualitative comparison of anytime interframe segmentation.** The top row establishes the visual context, displaying the input RGB frame at time $t$, the event stream from $t$ to $t + \delta t$, and the target **Ground Truth** (GT) segmentation at $t + \delta t$. The bottom row presents a zoomed-in comparison of the GT against the outputs of all evaluated methods.

evident across our benchmarks. While interpolation offers a modest improvement over the LFR (Baseline) on real-world datasets like DSEC and M3ED, it struggles significantly on synthetic data. On SHF-DSEC, it performs even worse than the LFR (Baseline) (55.89% vs. 61.73%), despite successful fine-tuning of the interpolation model (PSNR improved from 23.45 to 26.07). We observe a "PSNR-mIoU Paradox": improving photometric quality ($26.07 \to 27.43$ dB) via lower interpolation ratios paradoxically degrades semantic accuracy ($55.89\% \to 55.03\%$). This confirms an objective misalignment: reconstruction targets perceptual smoothness, often blurring discriminative boundaries. In contrast, our feature-space approach is inherently robust to the pixel-level micro-misalignments that plague image interpolation. The model's lack of robustness is further underscored on DSEC-Night, where the severe day-to-night domain shift renders the pre-trained interpolation model ineffective, making a meaningful comparison inapplicable. This demonstrates that our direct feature propagation is a more robust, practical, and causally-sound solution.

Finally, we compare against causal **LFR + Fusion** baselines (e.g., CMNeXt), which represent an alternative multi-modal approach. While this fusion strategy offers an improvement over the LFR (Baseline) in most real-world scenarios (Table 1), it still falls significantly short of our propagation-based framework. The performance gap is particularly pronounced on the high-dynamic **M3ED-Drone** dataset, where our method (64.28%) outperforms CMNeXt (59.56%) by a substantial margin of

**4.72% mIoU**. We conjecture that this stems from the inherent difficulty of direct fusion: the network must *implicitly* learn to align dense semantic features with sparse, texture-less event cues. This can be suboptimal, especially in high-motion scenes. In contrast, our framework's explicit, flow-guided *propagation* provides a stronger inductive bias for motion, geometrically warping features to maintain semantic consistency over time. This architectural advantage makes our approach fundamentally more effective for the anytime segmentation task.

**Qualitative Comparison.** Qualitatively, Fig. 4 provides compelling visual evidence for our method's superiority. We first observe the **LFR (Baseline)**, which fails to account for object/ego motion during the blind interval, resulting in a clear temporal misalignment or **"perceptual gap"** where segmented objects are visibly offset from their ground truth locations. This is particularly evident in the highly dynamic M3ED datasets. In contrast, while the **LFR Interpolation (TLX+Seg.)** method corrects for motion, it often produces **blurry and indistinct object boundaries**, an artifact of the image interpolation process that struggles to create photorealistic details. The **LFR Fusion (CMNeXt)** approach suffers from a different issue: by directly fusing sparse event features with dense image features, it can create **semantic ambiguity**, as seen in the M3ED-Quadruped example where object shapes are distorted.

Our method (**Ours**) overcomes all these limitations. It not only accurately compensates for the temporal gap, but also generates sharp and precise boundaries. This fine-grained accuracy is consistently demonstrated across datasets: on DSEC and M3ED, our method successfully delineates the challenging narrow gaps between pedestrians' legs, and on SHF-DSEC, it clearly separates the small figure of a person from their motorcycle. This ability to capture intricate detail while maintaining temporal consistency underscores the effectiveness of our explicit feature propagation framework.

**Anytime Performance and Robustness to Temporal Gaps.** We further analyze the *anytime* performance of all *causal* methods by evaluating their robustness to increasing temporal gaps ($\delta t$) on the high-frequency SHF-DSEC dataset, as visualized in Figure 5. Our framework (solid blue line) demonstrates exceptional stability, maintaining a consistently high mIoU across the entire 10–100 ms range, which showcases its true anytime capability. In stark contrast, the **LFR (Baseline)** (dark gray dashed line) suffers from a dramatic performance collapse, plummeting from 64.94% at 10ms to 58.80% at 100ms. This steep decline visually quantifies the severity of the "perceptual gap" for naive

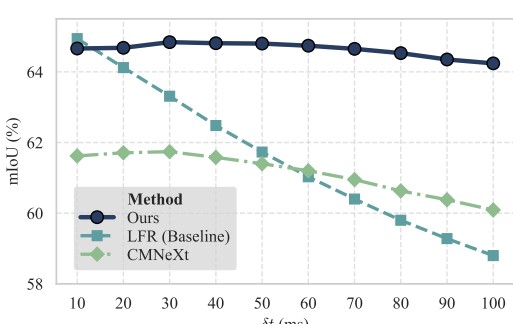

Figure 5: **Anytime performance on SHF-DSEC.** Our method (solid blue) remains stable, while baselines degrade as the temporal gap $\delta t$ increases.

LFR systems. Interestingly, the **LFR + Fusion** method (light green line) exhibits a more nuanced behavior: while starting with a lower mIoU than the LFR (Baseline), its degradation is less severe, leading to a crossover point at approximately $\delta t = 60$ ms. This suggests that simple fusion provides some resilience against temporal decay but is ultimately an insufficient and suboptimal solution. Our method is the only approach that remains robustly effective across all intervals, proving the superiority of our explicit propagation mechanism for bridging the blind time interval.

## 5.2 ABLATION STUDIES

To demonstrate the effectiveness and robustness of our proposed method, we conducted a series of ablation studies on the DSEC dataset. These studies analyze the impact of optical flow accuracy, the choice of warping method, and the contribution of the memory bank.

**Robustness of Pretrained Flow.** To verify that our model does not overfit to specific flow supervision, we conducted two cross-domain experiments. First, our method achieves state-of-the-art performance on the unseen M3ED dataset using a flow estimator pretrained only on DSEC, demonstrating strong zero-shot transferability. Second, replacing the DSEC-pretrained flow network (Prophesee sensors / 640x440) with one trained on the distinct MVSEC Zhu et al. (2018) dataset (DAVIS346 / 346x260, small displacements) results in a negligible 0.14% mIoU drop on DSEC. These results confirm that our framework learns robust motion representations and generalizes well across different domains.

**Robustness to Different Flow Estimators.** Our framework demonstrates strong robustness when paired with different optical flow estimation methods. In Table 2, we compare the segmentation performance (mIoU%) of our framework when equipped with different optical flow estimators, including the image-based RAFT Teed & Deng (2020) and several event-based methods such as bflow Gehrig et al. (2024), IDNet Wu et al. (2024), and E-RAFT Gehrig et al. (2021b). The competing paradigms of interpolation and direct fusion (detailed in §5.1) are significantly outperformed by our propagation-based approach, achieving suboptimal mIoU scores of 70.38% and 70.13%, respectively. In contrast, **our method consistently achieves much higher performance** across all tested flow estimators. Notably, our framework

Table 2: Ablation of different paradigms and their specific implementations on DSEC-Semantic. *Indicates inputs adapted for the anytime task.

| Method / Paradigm | mIoU (%) |
|---|---|
| **LFR + Interpolation (Fig. 3b)** | |
| TLX + Seg. | 68.17 |
| **LFR + Fusion (Fig. 3c*)** | |
| CMNeXt | 70.13 |
| **Ours (LFR + Propagation) (Fig. 3d)** | |
| w/ RAFT | 72.93 |
| w/ bflow | 73.38 |
| w/ IDNet (iter 4, 1/8) | 73.00 |
| w/ IDNet (iter 4, 1/4) | 73.62 |
| w/ E-RAFT-Lite (iter 12, 1/16) | 73.49 |
| w/ E-RAFT | **73.82** |

demonstrates robustness to lower-quality flow estimates, achieving a strong **73.49% mIoU** even when using a coarse flow map from E-Raft-Lite version flow estimator generated with only 1/16th resolution. This demonstrates that our method is largely **agnostic to the specific choice of flow estimator** and does not rely on highly precise flow. This robustness stems from two key design elements: (1) **Uncertainty-aware warping** down-weights unreliable motion regions, limiting the negative impact of imperfect flow; (2) The **temporal memory module** provides long-term context to correct local misalignments. Together, these components ensure reliable temporal propagation and stable segmentation performance, even with imperfect flow inputs.

**Ablation of Uncertainty Map** To understand the empirical behavior of the uncertainty-aware warping mechanism, we visualize the learned Uncertainty Map ($S$) alongside the input Event Voxel and Estimated Flow in Figure 6. The visualization confirms that the ScoreNet effectively acts as a reliability filter by measuring the consensus between the two modalities:

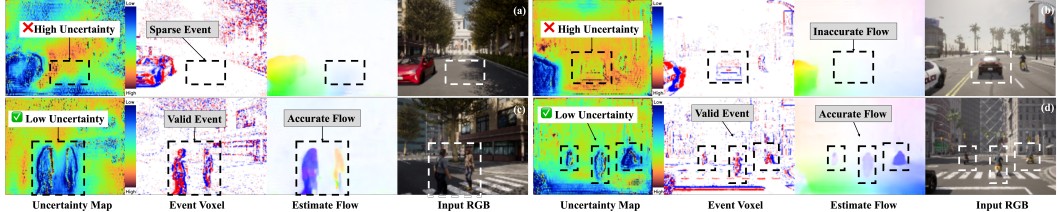

Figure 6: Visualization of Uncertainty Map behavior. (a) **Sparsity:** Flow unsupported by sparse events triggers high uncertainty, suppressing hallucinations. (b) **Inaccuracy:** Disagreement between event edges and inaccurate flow triggers high uncertainty, filtering errors. (c)(d) **Alignment:** Consistent flow-event alignment (e.g., pedestrians, riders) yields high confidence for propagation.

Table 3: Impact of ScoreNet.

| Method | DSEC | SHF | Night |
|---|---|---|---|
| w/o Score | 72.74 | 63.31 | 41.46 |
| **Ours** | **73.82** | **64.80** | **41.86** |

We validate the uncertainty module quantitatively in Table 3. Incorporating the learned confidence map consistently improves performance across all benchmarks (e.g., +1.08% on DSEC), confirming its effectiveness in filtering unreliable motion cues caused by noise or sparsity.

**Ablation on the Warping Domain.** To effectively propagate information over time, the choice of what data to warp, the warping domain, is a critical design decision. We conducted an ablation study to compare our proposed feature-level warping against two common alternatives: warping the raw input images (Image Warping) and warping the final segmentation predictions (Segmentation Warping). As detailed in Table 4, our feature warping strategy achieves a state-of-the-art 73.82% mIoU. This result significantly surpasses warping at the image level (72.37%) and the prediction level (71.63%). Furthermore, all methods employing explicit motion compensation (warping) show a distinct advantage over the simple interpolation baseline (70.38%). These results provide clear evidence for our central hypothesis: propagating rich, deep semantic features through motion-compensated alignment is the most effective strategy for maintaining high-quality, temporally consistent results in an anytime segmentation task.

Table 4: Different warping strategies on DSEC ($\delta t = 50$ ms).

| Method | mIoU (%) |
|---|---|
| Image Interpolation | 70.38 |
| Image Warping | 72.37 |
| Segmentation Warping | 71.63 |
| **Feature Warping (Ours)** | **73.82** |

Table 5: The Effectiveness of the Memory Module Over Long Temporal Gaps on DSEC (mIoU %)

| Method / $\delta t$ (ms) | 50 | 200 | 400 | 800 |
|---|---|---|---|---|
| Lower Bound | 67.67 | 57.06 | 51.18 | 45.34 |
| Ours (w/o Mem) | 73.49 | 72.00 | 67.02 | 57.33 |
| **Ours (w/ Mem)** | **73.82** | **72.72** | **68.60** | **59.55** |

**Influence of Memory.** The effectiveness of our long-term memory module in preserving temporal consistency is systematically evaluated over increasingly long temporal intervals, up to 800 ms, on the DSEC dataset. As shown in Table 5, the performance gap between the model with and without the memory module remains small at a short interval of 50 ms, with mIoU improving only slightly from 73.49% to 73.82%. However, as the temporal gap increases, the benefits of the memory module become increasingly significant. At 200 ms, the model equipped with memory achieves an mIoU of 72.72%, outperforming the counterpart without memory by 0.72 percentage points. This trend continues at 400 ms, where the improvement grows to 1.58 percentage points, demonstrating the module's ability to retain semantic information over time. Most notably, at the longest interval of 800 ms, the model with memory reaches an mIoU of 59.55%, surpassing both the lower bound by a large margin and the memory-free variant by 2.22 percentage points—nearly eight times the initial gain observed at 50 ms. These results clearly illustrate that the memory module plays a critical role in mitigating feature decay and maintaining robust temporal alignment, especially in challenging scenarios with sparse or infrequent RGB observations. Its contribution is not merely incremental but becomes indispensable as temporal continuity is increasingly disrupted.

## 6 CONCLUSION

In this work, we addressed the critical problem of "perceptual gaps" that plague standard low-frame-rate (LFR) systems in dynamic environments. We introduced and formalized a new task, **Anytime Interframe Semantic Segmentation**, and proposed **LiFR-Seg**, a novel framework that effectively bridges these gaps. Our approach propagates rich semantic information from a single RGB frame forward in time, guided by an event-driven motion field. The core of our method lies in an uncertainty-aware feature warping mechanism that robustly handles noisy motion estimates, and a temporal memory module that ensures coherence in highly dynamic scenes.

Our extensive experiments provide compelling evidence for the efficacy of this paradigm. We demonstrated that our LFR system achieves performance remarkably comparable to an ideal high-frame-rate (HFR) upper bound, closing the performance gap to less than 0.09% on the DSEC dataset. Furthermore, we validated the extreme robustness of our framework: in highly dynamic M3ED tests, our method closely matches the HFR baseline, while in a challenging zero-shot test on DSEC-Night, it even surpasses the RGB-based upper bound, proving its viability where traditional cameras fail. We believe this work presents a significant step towards a new paradigm of efficient and reliable perception. The principles of event-guided propagation demonstrated here can be extended to other dense prediction tasks, such as depth or flow estimation. Ultimately, this works showcases a promising path towards decoupling perceptual frequency from sensor hardware limitations, paving the way for more ubiquitous and reliable autonomy.

## 7 ACKNOWLEDGEMENTS

The work has been supported by Hong Kong Research Grant Council - General Research Fund Scheme (Grant No. 17202422, 17212923, 17215025) Theme-based Research (Grant No. T45-701/22-R), Strategic Topics Grant (Grant No. STG3/E-605/25-N), and Voyager Research, Didi chuxing. Part of the described research work is conducted in the JC STEM Lab of Robotics for Soft Materials funded by The Hong Kong Jockey Club Charities Trust. This research was partially conducted by ACCESS – AI Chip Center for Emerging Smart Systems, supported by the InnoHK initiative of the Innovation and Technology Commission of the Hong Kong Special Administrative Region Government.

## 8 ETHICS STATEMENT

We confirm adherence to the ICLR Code of Ethics and have carefully evaluated the ethical implications of our research. We present our key considerations below.

1. **Applications and Responsible Use**

   Our work advances perception ability in real scenarios, aiming to improve scene understanding and safety in transportation systems. We acknowledge that perception technologies may have applications beyond our intended scope. We encourage the responsible deployment of our methods in accordance with applicable regulations and safety standards for autonomous systems development.

2. **Data Handling and Compliance**

   We utilize established public datasets (DSEC, M3ED) under their respective licensing agreements. Furthermore, we also leverage Carla to generate a synthetic dataset, SHF-DSEC, which only contains a virtual environment and identity. These datasets contain anonymized sequences without personal identifiers. Our research strictly follows the data usage policies established by the dataset providers and does not involve additional data collection or processing of sensitive information.

## 9 REPRODUCIBILITY STATEMENT

To ensure reproducibility of our results, we have provided comprehensive details necessary to replicate our experiments. The main text outlines our experimental settings in Section C.1, including dataset usage, evaluation metrics, and training configurations. Further implementation specifics are documented in Appendix C.1, which covers network architecture details, hyperparameter settings, and the use of software libraries. All experiments are based on publicly available datasets, including the DSEC and the M3ED dataset, and the self-created dataset SHF-DSEC, and use clearly defined data splits and evaluation protocols consistent with prior work. To further support the research community, we commit to releasing our full source code and preprocessed datasets upon acceptance of this paper.

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

## A  BENCHMARK INFORMATION

Here is the detailed information on each dataset.

**DSEC.** The real-world **DSEC** dataset Gehrig et al. (2021a) provides 11 classes of segmentation pseudo-labels at a low frequency of **20 Hz**, alongside a high-resolution event stream. This restricts our primary evaluation to a temporal gap of $\delta t = 50$ ms.

**M3ED.** The **M3ED** dataset Chaney et al. (2023) is used to evaluate robustness to diverse dynamics, providing RGB frames at a **25 Hz** frequency. We specifically test on its challenging **Drone** and **Quadruped** splits, which feature rapid ego-motion.

**DSEC-Night.** The **DSEC-Night** benchmark Xia et al. (2023) is an **evaluation-only** set of 150 manually annotated nighttime labels. It serves as a rigorous **zero-shot** test for generalization to extremely low-light conditions.

**SHF-DSEC.** To overcome the 20 Hz limitation of DSEC for evaluating anytime performance, we introduce our synthetic **SHF-DSEC** dataset, built with the CARLA simulator Dosovitskiy et al. (2017). This dataset provides dense, ground-truth segmentation maps for 11 classes, synchronized with RGB frames and event streams, all at a high frequency of **100 Hz**. The training set (1,260 samples) is drawn from CARLA towns 01-05, while the test set (180 samples) uses town 10 to evaluate domain generalization. Event streams are simulated based on logarithmic intensity changes. The 100 Hz ground truth is crucial, as it allows us to rigorously evaluate our model's "anytime" capability at much finer temporal intervals, such as $\delta t = 10$ ms.

## B  DATA GENERATION IN SHF-DSEC

The SHF-DSEC dataset is sampled at 10 ms intervals, comprising a total of 16,200 samples per sequence. It integrates data from three synchronized sensors: an RGB camera, an event camera based on the configuration by Hidalgo et al. Javier Hidalgo-Carrio & Scaramuzza (2020), and a segmentation camera adhering to the DSEC setup by Gehrig et al. Gehrig et al. (2021a). All sensors operate at a resolution of $480 \times 640$ with a field of view of $57.5°$, ensuring consistent and high-quality inputs for dynamic scene segmentation tasks. As detailed in Table 6, the dataset is specifically designed to enhance segmentation variety and robustness for event-based methods, featuring six distinct simulated environments generated using the CARLA simulator. These environments reflect diverse urban and natural settings, with vehicle and pedestrian configurations inspired by Aliminati et al. Aliminati et al. (2024).

The SHF-DSEC dataset is synthesized within the CARLA simulator, rendering high-fidelity scene frames under varying lighting and motion conditions. To faithfully emulate real sensors, we configured the simulation with a rigorous 1 ms fixed time-step (1000 Hz). This high-frequency physical sampling, rather than linear interpolation, accurately captures rapid inter-frame dynamics without temporal aliasing. An event $e = (x, y, t, \text{pol})$ is triggered at pixel $(x, y)$ and timestamp $t$ when the change in logarithmic intensity $L(x, y, t)$ exceeds a predefined threshold. Specifically, an event occurs if $|L(x, y, t) - L(x, y, t - \delta t)| = \text{pol} \cdot C$, where $C = 0.3$ is the contrast threshold, $\delta t$ denotes the time elapsed since the last event at that pixel, and $\text{pol} \in \{+1, -1\}$ represents the polarity indicating a brightness increase or decrease, respectively. This mechanism generates a realistic event stream that effectively captures dynamic scene changes.

The dataset encompasses 11 annotation classes for segmentation: background, building, fence, person, pole, road, sidewalk, vegetation, car, road lines, and traffic sign. Notably, the "wall" class from the original DSEC dataset was replaced with "road lines" due to the visual similarity between simulated walls and buildings in CARLA, which poses challenges for accurate differentiation.

Table 6: SHF-DSEC Dataset Structure

| Map | Description | Sequences | Usage |
|-----|-------------|-----------|-------|
| Town01 | Small town featuring a river and bridges | 1 | Training |
| Town02 | Small town with a mix of residential and commercial buildings | 2 | Training |
| Town03 | Larger urban setting with a roundabout and multiple junctions | 2 | Training |
| Town04 | Mountainous town with an infinite highway | 1 | Training |
| Town05 | Grid-based town with cross-junctions, a bridge, and multi-lane directions | 1 | Training |
| Town10HD_Opt | Downtown area with skyscrapers, residential buildings, and an ocean promenade | 1 | Testing |

## C   EXPERIMENTAL SETUP

### C.1   TRAINING AND IMPLEMENTATION DETAILS

**Training Strategy.**   Our entire framework is trained end-to-end using the OhemCrossEntropy loss Shrivastava et al. (2016) to mitigate class imbalance. A key aspect of our training is that supervision is applied only at the discrete RGB frame timestamps $t+\Delta t$, where ground truth $\text{Seg}_{t+\Delta t}$ is available. To achieve this, the feature $F_{t+\delta t}$, which has been propagated to an intermediate time (typically the midpoint, $\delta t = \Delta t/2$), is warped a second time to $F_{t+\Delta t}$ before being passed to the final segmentation decoder. Specifically, this second warp utilizes a new motion field $\hat{M}_{t+\delta t \rightarrow t+\Delta t}$, which is estimated by feeding event slice $E_{t \rightarrow t+\delta t}$ and $E_{t+\delta t \rightarrow t+\Delta t}$ into the flow estimator. This ensures that the entire propagation chain is differentiable and aligned with the available supervision.

**Hyperparameters.**   We use the AdamW optimizer Kingma (2014) with a learning rate of $1e^{-4}$ and a weight decay of $5e^{-3}$. A polynomial decay schedule is employed for the learning rate, with a 10-epoch warm-up phase followed by decay with a power of 0.95. All models were trained on two NVIDIA RTX 4090 GPUs for 200 epochs until convergence, using a total batch size of 4.

**Anytime Inference.**   At test time, our framework's "anytime" capability is realized. By providing the relevant event slice $\mathcal{E}_{t-\Delta t \rightarrow t+\delta t}$ for *any* target time $\delta t$, our model can compute the corresponding motion field and propagate features on the fly. This enables dense segmentation at arbitrary temporal resolutions without any modification or retraining of the model.

### C.2   EXPERIMENTAL SETUP

To rigorously validate our framework, we conducted extensive evaluations on the real-world DSEC and M3ED datasets, our synthetic SHF-DSEC dataset, and the DSEC-Night benchmark. For a fair comparison, all methods leverage the same **Segformer-B2** backbone and are trained to convergence on their respective datasets, with the exception of the zero-shot DSEC-Night evaluation. We established four categories of external baselines: the **HFR Upper Bound** (an ideal system with $I_{t+\delta t}$), the **LFR Lower Bound** (a naive approach using only $I_t$), **Interpolation Baselines** (e.g., TLX + Segformer), and **Multi-Modal Fusion Baselines** (e.g., CMNeXt).

Our framework employs the SegFormer model with the MiT-B2 backbone Xie et al. (2021), which utilizes a hierarchical Transformer encoder and a lightweight MLP decoder to generate dense semantic predictions. The model is initialized with weights pre-trained on ImageNet and then fine-tuned on the DSEC and SHF-DSEC datasets. To ensure a fair and consistent comparison, all baseline methods, including, TimeLens-XL Ma et al. (2024), EISNet Xie et al. (2024), and CMNeXt Zhang et al. (2023), were also re-trained from scratch on both datasets until convergence. We made a specific adaptation for TimeLens-XL due to its architectural constraints, which require input dimensions to be multiples of 32. For this baseline, we applied center cropping to our standard $440 \times 640$ input, resulting in a resolution of $384 \times 608$. Accordingly, its evaluation was performed by comparing predictions against

ground truth segmentation maps cropped to the same resolution, ensuring an unbiased assessment across all methods.

## D  IMPRACTICALITY OF THE RGB-BASED HFR SYSTEM

Table 7: Event camera vs. high-speed RGB camera.

|  | **Event Camera** | **High-Speed RGB** |
|---|---|---|
| **Camera Type** | Prophesee EVK4 HD | Phantom MTX-7510 |
| **Resolution** | 1280×720 | 1280×640 |
| **Max FPS** | >1M | 94K |
| **Price (USD)** | $5K | $150K |
| **Power (W)** | 1.5 | >325 |
| **Dynamic Range (dB)** | >120 | 51 |

We believe that our proposed problem and solution hold high practical value. Below, we compare our approach with the combination of a high-speed RGB camera and SegFormer in terms of hardware overhead.

As shown in Table 7, event cameras feature extremely high temporal resolution (Prophesee, 1280x720, >1M FPS) at a low price ($5kUSD), small power consumption (1.5W), and large dynamic range (>120db). In comparison, a high-speed RGB camera (Phantom MTX-7510, 1280x640, 94K FPS) has a high price ($150kUSD), large power consumption (>325W), and small dynamic range (51db).

## E  COMPUTATIONAL EFFICIENCY ANALYSIS

We analyze the computational efficiency of LiFR-Seg on an NVIDIA RTX 3090 at $440 \times 640$ resolution. The core efficiency of our paradigm stems from **amortization**: the heavy Image Encoder is executed only once per keyframe ($I_t$), while subsequent predictions ($t + \delta t$) utilize lightweight propagation modules. Theoretically, for $N$ propagated frames, the cost saving is proportional to $N \times (C_{\text{encoder}} - C_{\text{modules}})$. In our practical implementation, we utilize a lightweight variant (LiFR-Seg-Lite) with a RAFT-small backbone and $1/16$ flow resolution, which maintains a competitive 73.49 mIoU (vs. 73.82 mIoU for the full version) while ensuring minimal computational overhead.

Table 8: Computational cost comparison on RTX 3090. LiFR-Seg-Lite achieves the lowest amortized FLOPs while maintaining real-time speeds.

| **Method** | **Params (M)** | **Avg Cost ($N = 1$)** | | **Avg Cost ($N = 10$)** | |
|---|---|---|---|---|---|
| | | **GFLOPs** | **FPS** | **GFLOPs** | **FPS** |
| HFR Upper Bound | 25.8 | 42.04 | 72.8 | 42.04 | 72.8 |
| LFR + Fusion (EISNet) | 34.5 | 72.73 | 34.7 | 72.73 | 34.7 |
| LFR + Fusion (CMNeXt) | 58.7 | 68.12 | 29.1 | 68.12 | 29.1 |
| LFR + Interp. (TLX) | 33.2 | 200.77 | 29.5 | 248.51 | 26.7 |
| **Ours (LiFR-Seg-Lite)** | 30.7 | **40.43** | **65.6** | **38.89** | **60.3** |

As shown in Table 8, our amortized cost (40.43 GFLOPs) is lower than the HFR baseline (42.04 GFLOPs) and significantly outperforms fusion (>68 GFLOPs) and interpolation (>200 GFLOPs) methods. Although our latency (65.6 FPS) is slightly higher than HFR due to the memory-bandwidth bottleneck of correlation lookups, it remains well within real-time requirements. Furthermore, this memory-bound characteristic suggests strong scaling potential on modern high-bandwidth hardware (e.g., A100 or RTX 40-series), highlighting its viability for high-speed deployment in autonomous systems.

## F    LIMITATIONS

Despite these promising results, our current approach has several limitations. While LiFR-Seg proves robust to high-speed ego-motion (as seen on M3ED), our current evaluation includes limited datasets featuring high-speed object dynamics (e.g., extreme localized motion blur, or highly non-linear motion). Challenges such as complex non-linear deformations (e.g., sports) or severe source-frame motion blur represent a valuable frontier for future propagation-based research. To overcome these challenges, our immediate next step will involve constructing new real-world and synthetic datasets explicitly designed to incorporate high-speed object motion scenarios. Developing these datasets will enable comprehensive validation of our approach under more demanding and realistic conditions, expanding its applicability to critical domains such as autonomous driving, sports analytics, and drone navigation. Furthermore, integrating our anytime segmentation framework with advanced, specialized hardware platforms optimized for event-driven computation could significantly enhance real-time processing efficiency. This combined hardware-software co-design would be particularly beneficial for resource-constrained edge devices, enabling robust, high-temporal-resolution segmentation in practical, real-world scenarios.

## G    BROADER IMPACTS

To enhance our model, we plan to adapt it for streaming inputs in an online fashion. By utilizing the optical flow obtained from the previous time step as an initialization for the next time step's flow estimation, we can further improve computational efficiency. Additionally, we aim to extend our research into the broader domain of video-based dynamic segmentation. Video segmentation introduces challenges such as variable frame rates, diverse lighting conditions, and persistent occlusions. We are confident that our framework can be enhanced to address these complexities, thereby expanding its real-world applications and significantly advancing the state-of-the-art in dynamic semantic segmentation.

## H    DECLARATION OF LLM USAGE

Large Language Models (LLMs) were used to assist in language editing, grammar refinement, and improving the overall clarity and readability of the manuscript. However, all scientific ideas, methodologies, experimental designs, data analysis, and conclusions presented in this work are entirely the product of the authors' independent research and intellectual effort. The authors have carefully reviewed, revised, and approved all content and take full responsibility for the accuracy, integrity, and authenticity of the work.

