# OpenReview forum: "LiFR-Seg: Anytime High-Frame-Rate Segmentation via Event-Guided Propagation"
_ICLR.cc/2026/Conference — ICLR 2026 Poster_

### Official Review · Reviewer_xGY9 · 2025-10-22

**Soundness:** 3
**Presentation:** 3
**Contribution:** 3
**Rating:** 4
**Confidence:** 3

**Summary:**

The paper proposes a novel framework for Anytime Semantic Segmentation to address the challenges of high-temporal-resolution segmentation in dynamic environments by integrating RGB images and event data. Leveraging the complementary strengths of RGB frames (spatially dense but temporally sparse) and event streams (temporally dense but spatially sparse), their method proposes uncertainty-aware optical flow estimation, learnable feature warping, and a memory mechanism to align and enhance features for segmentation at arbitrary time points. The framework is evaluated on the DSEC and a newly introduced synthetic dataset, SHF-DSEC, demonstrating better performance and robustness. However, they didn't elaborate on training time and how fast their model run compared to SOTA methods.

**Strengths:**

•	The design of the framework is relatively well explained, incorporating uncertainty-aware optical flow estimation, learnable feature warping to achieve precise feature alignment across modalities
•	The proposed synthetic dataset (SHF-DSEC) is a valuable contribution to the field for it provides a high-temporal-resolution segmentation dataset

**Weaknesses:**

•	Didn’t elaborate on training time and number of parameters of the proposed model against baselines they compare with
•	Did they explain how they generate event streams in SHF-DSEC
•	No report on how fast their model runs compared to other schemes.

**Questions:**

The optical flow estimator, E-Raft, is pretrained on DSEC and SHF-DSEC using their corresponding ground truth optical flow. This setup may lead to unfair evaluation because the model could overfit to the ground truth flow of the corresponding dataset. It is unclear whether the trained model can generalize to unseen data.

---

> ### Author Response · Authors · 2025-11-21
> **Official Response to Reviewer xGY9 Part 1: W1-W3 Clarification**
>
> We thank the reviewer for recognizing the value of our framework and dataset. We have addressed your feedback regarding efficiency and generalization with detailed experiments below.
> >**W1:** Didn’t elaborate on training time and number of parameters...
>
> **A:** We provide the requested comparison (Table below) to demonstrate our method's cost-effectiveness on DSEC (2x RTX 3090).
>
> |**Method**|**Paradigm**|**Params (M)**|**Training Time**|**mIoU (%)**|
> |:-:|:-:|:-:|:-:|:-:|
> |**HFR (Upper Bound)**|*Ideal HFR RGB*|25.8|~2.0h|73.91|
> |LFR (Baseline)|*RGB only*|25.8|~2.0h|67.67|
> |LFR + Interp. (TLX)|*Interpolation*|33.2|~8.0h|68.17|
> |LFR + Fusion (CMNeXt)|*Direct Fusion*|58.7|~3.0h|70.13|
> |**Ours (LiFR-Seg)**|*Propagation*|**30.7**|**~3.0h**|73.82|
> |**Ours (w/ Pre-train)**|*Transfer Learning*|**30.7**|**~20 min**|**74.01**|
>
> **Efficiency:** We introduce minimal parameters (**+4.9M** vs. Baseline), significantly outperforming the bloated Fusion baseline (CMNeXt, **58.7M**) while training $\sim$ 2.6x faster than Interpolation (TLX).
>
> **Performance:** Standard LiFR-Seg achieves a massive gain (**+6.15%**) over the LFR Baseline. While HFR is efficient, it requires prohibitive high-bandwidth hardware; LiFR-Seg effectively matches this "Ideal Bound" (gap < **0.09%**) using standard LFR sensors.
> **Sim-to-Real Benefit:** Leveraging our SHF-DSEC dataset for pre-training allows for rapid convergence. Fine-tuning for just 25 epochs (~20 min) yields **74.01% mIoU**, surpassing both our previous best and the HFR Upper Bound (73.91%).
> >
>
> >**W2:** Did they explain how they generate event streams in SHF-DSEC?
>
> **A:** We respectfully note that the generation details were explicitly included in **Appendix B** of the original submission. To clarify the generation mechanism:
> 1. **Theoretical Model:** We utilize the standard generative model where log-intensity changes trigger asynchronous events: $L(x,y,t) - L(x,y,t-\delta t) = pol \cdot C$ .
> 2. **High-Fidelity Sampling:** To capture inter-frame dynamics, we configured CARLA with a rigorous **1 ms fixed time-step (1000 Hz)**. Unlike linear interpolation, this uniform physical sampling accurately captures high-speed intensity changes without temporal aliasing.
> 3. **Configuration:** We set the contrast threshold $C=0.3$ to match the signal-to-noise ratio and sparsity of commercial sensors (e.g., Prophesee Gen3).
>
> We have updated **Appendix B** to explicitly reference these details and thank the reviewer for the suggestion.
>
> >**W3:** No report on how fast their model runs compared to other schemes.
>
> **A:** We respectfully refer the reviewer to the **General Response** for a detailed breakdown. In summary, LiFR-Seg achieves **~65.6 FPS** (RTX 3090), running **nearly $2\times$ faster** than Fusion (34.7 FPS) and Interpolation (29.5 FPS) baselines. These results confirm our framework's high efficiency and strong potential for practical deployment.

---

> ### Author Response · Authors · 2025-11-22
> **Official Response to Reviewer xGY9 Part 2: Q1 Optical Flow Generalization**
>
> >**Q1:** The optical flow estimator, E-Raft, is pretrained on DSEC... This setup may lead to unfair evaluation because the model could overfit... It is unclear whether the trained model can generalize to unseen data.
>
> **A:** Thank you for raising this important concern about potential overfitting of the optical flow estimator and the fairness of the evaluation! We fully agree that demonstrating cross-domain generalization is essential. To directly address this, we performed two complementary experiments showing that LiFR-Seg relies on **robust, transferable motion representations**, not dataset-specific flow supervision.
>
> **1. Generalization to Unseen Domains (DSEC $\to$ M3ED)**
>
> In our main experiments (Table 1), the optical flow estimator used for M3ED evaluation is **pretrained exclusively on DSEC**, because M3ED provides **no** flow ground truth.
>
> Despite substantial domain differences:
> - **DSEC:** Automotive driving, mostly planar motion
> - **M3ED:** Robotics, aggressive 6-DoF ego-motion with rapid rotations and large parallax
>
> LiFR-Seg still achieves strong performance:
> - **64.28% mIoU** on M3ED-Drone
> - **Close to HFR upper bound (difference of only 0.29%)**
> - **+9.05% improvement** over the LFR baseline
> This demonstrates that the pretrained flow network generalizes well to unseen sensors, unseen motion patterns, and unseen environments, indicating that the system does **not** overfit to DSEC-specific flow statistics.
>
> **2. Flow Pretrained on a Completely Different Dataset: MVSEC $\to$ DSEC**
>
> To directly test whether using DSEC flow ground truth gives an unfair advantage, we replaced the flow estimator with one pretrained on **MVSEC**, which differs drastically from DSEC:
>
> **Table: Cross-Domain Flow Generalization Analysis**
>
> |**Flow Source**|**Sensor / Res.**|**Flow Characteristics**|**DSEC mIoU (%)**|
> |:-:|:-:|:-:|:-:|
> |**DSEC (Original)**|Prophesee / 640x440|Large displacements, Driving scenarios|**73.82**|
> |**MVSEC (Cross-Domain)**|DAVIS346 / 346x260|Small displacements, Perfect alignment|**73.68**|
>
> Even though the MVSEC-pretrained flow:
> - comes from a different sensor,
> - has mismatched resolution,
> - contains much smaller and cleaner motions,
> the final DSEC performance closely matches the original performance.
>
> This negligible difference confirms that LiFR-Seg does not depend on memorizing dataset-specific flow patterns.
>
> These two experiments, **DSEC $\to$ M3ED** generalization and **MVSEC $\to$ DSEC** cross-domain transfer, demonstrate that:
> - LiFR-Seg does not overfit to dataset-specific flow ground truth,
> - the system generalizes robustly across sensors and motion regimes, and
> - our evaluation is fair and not inflated by in-domain flow training.
>
> We have incorporated this clarification into **Section 5.2 (Ablation Studies: Robustness of Pretrained Flow)** of the revised manuscript. We appreciate the reviewer for encouraging us to strengthen this aspect of the paper!

---

> > ### Comment · Reviewer_xGY9 · 2025-11-24
> >
> > Based on what I recalled, EISNet can achieve 74 mIoU. It runs with a latency of about 84 ms (close to 10 FPS) but your approach requires both Event encoder and flow encoder. So, how can your approach runs 6 times faster?

---

> > > ### Comment · Reviewer_xGY9 · 2025-11-24
> > >
> > > Did you use the whole EISNet mode or merely use the later part of EISNet when you run with EISNet? EISNet main contribution is to design event representations so if you remove the earlier part, the comparision is not too fair.

---

> > > ### Author Response · Authors · 2025-11-24
> > > **Response to Reviewer xGY9: Comparison with EISNet**
> > >
> > > >**Q:** Based on what I recalled, EISNet can achieve 74 mIoU... runs with a latency of about 84 ms... how can your approach runs 6 times faster?
> > >
> > > **A:** Thank you for the opportunity to clarify these metrics! We re-checked the original EISNet paper, which reports **73.07% mIoU** on DSEC ([Paper Table 1](https://ieeexplore.ieee.org/document/10477577), [Github README](https://github.com/bochenxie/EISNet) ) and does not provide specific latency figures for this dataset. In our standardized benchmarking on an NVIDIA RTX 3090 ($440 \times 640$), as mentioned in **General Response Quantitative Analysis**, we measured EISNet at **34.7 FPS**, which is faster than the recalled 10 FPS. Below, we provide a detailed analysis of the performance and latency differences under these standardized conditions.
> > >
> > > **1. Performance-Wise**
> > > The performance gap stems from the difference between the task EISNet was designed for and the "Anytime" task we evaluate.
> > > - **Standard Fusion Setting (EISNet Original):** The 73.07% mIoU reported in the original EISNet paper is evaluated on the **Standard Fusion** task: fusing an RGB image $I_t$ with *past* events $E_{t-\Delta t \to t}$ to segment the *current* frame $t$.
> > > - **Our Setting (Anytime Prediction):** As defined in **Figure 3c** and **Lines 269-284** of our paper (original submitted version), our task is to predict a *future* state $t+\delta t$ using $I_t$ and *forward-looking* events $E_{t \to t+\delta t}$. This is a harder predictive problem.
> > >
> > > |**Setting**|**Task Nature**|**Input Data**|**Target Output**|**DSEC mIoU**|
> > > |:-:|:-:|:-:|:-:|:-:|
> > > |**EISNet Original**|Enhancement|$I_t$ + Past Events ($E_{t-\Delta t \to t}$)|**Current** ($Seg_t$)|**73.07%**|
> > > |**Our Adaptation**|Prediction|$I_t$ + Future Events ($E_{t \to t+\delta t}$)|**Future** ($Seg_{t+\delta t}$)|**68.11%**|
> > >
> > > We adapted EISNet to this predictive setting. While its performance drops to **68.11%** due to the increased difficulty of predicting the future rather than enhancing the present, it still correctly outperforms the RGB-only baseline (67.67%). The comparison is fair within the context of the Anytime task.
> > >
> > > **2. Latency-Wise**
> > > Since latency was not reported for DSEC in the original work, we benchmarked both methods on an NVIDIA RTX 3090 with $440 \times 640$ input:
> > > - **EISNet:** Measured at **34.7 FPS**.
> > > - Ours: Measured at 65.6 FPS.
> > > 	While EISNet is faster in our measurements than the 10 FPS recalled, our method still achieves a nearly $2\times$ speedup.
> > >
> > > **3. Source of Efficiency**
> > > The observed efficiency advantage derives from architectural choices driven by distinct design philosophies. As a fusion-based method, EISNet employs a relatively heavy **MiT-B0 Transformer** backbone organized in **four hierarchical stages** ($\sim$3.54 GFLOPs) to extract rich semantic representations from events, which is essential for effective cross-modal fusion. In contrast, the event and flow encoders within our Splatting Module serve a targeted geometric purpose: computing an **uncertainty map** to handle event sparsity and flow-event misalignment for robust feature warping. This focused objective allows the use of lightweight, shallow **Conv2d layers** ($\sim$7.60 MFLOPs). Thus, the significant difference in encoding cost reflects the distinct requirements of semantic feature extraction versus geometric motion guidance.

---

> > > > ### Comment · Reviewer_xGY9 · 2025-11-25
> > > >
> > > > I still believe that this is not a fair comparison. Your adapted EISNET is no longer EISNet for their paper main contribution is to design better event representations.
> > > > Your approach relies on future events and hence it can't be utilized in real settings.
> > > > Do you have  latency contribution of individual components of your model e.g. how long it takes for flow encoder, how long it takes for event encoder?
> > > > Do you have any latency log file for your expt?

---

> > > > > ### Author Response · Authors · 2025-11-26
> > > > > **Official Response to Reviewer xGY9: Followup Comment 4 Fairness of EISNet Baseline**
> > > > >
> > > > > >**Q:**: I still believe that this is not a fair comparison. Your adapted EISNET is no longer EISNet for their paper main contribution is to design better event representations.
> > > > >
> > > > >
> > > > > **A:** We thank the reviewer for raising this concern. We would like to clarify that in our implementation, we **fully preserved the original EISNet architecture**, including its core contribution, the AEIM/AET event representation module. Our adaptation only involved modifying the input event interval (using $E_{t+\delta t}$ instead of $E_{t-\Delta t}$) and target supervision (predicting $Seg_{t+\delta t}$ instead of $Seg_t$) to match our evaluation protocol, but the architectural design and the event-representation mechanism remained exactly as proposed in the original paper. We hope this clarification alleviates the concern and reassures the reviewer that our comparison was conducted fairly and with full respect to EISNet’s intended design.

---

> ### Author Response · Authors · 2025-11-24
> **Response to Reviewer xGY9: Fairness of EISNet Baseline**
>
> >**Q:** Did you use the whole EISNet model or merely use the later part... if you remove the earlier part, the comparison is not too fair.
>
> **A:** Thank you for this scrutiny regarding baseline fairness. We confirm that we implemented the **full EISNet architecture**, explicitly including its core **Activity-Aware Event Integration Module (AEIM)** for its event representation **Activity-Aware Event Tensor (AET)**.

---

> ### Author Response · Authors · 2025-11-24
> **Response to Reviewer xGY9: Generation of HFR Upper Bound**
>
> > **Q:** Can you provide more details on how you generate the HFR upper bound results?
>
> **A:** We noticed an additional comment regarding the HFR generation posted in a different thread. For your convenience and to maintain a centralized discussion, we address that question here:
>
> We thank the reviewer for this question. As defined in Lines 264-265 (original submitted version), the HFR Upper Bound represents an ideal system that has access to the privileged target RGB frame $I_{t+\delta t}$ (which is physically unavailable to the LFR system during the blind interval).
> The pipeline is a standard image segmentation inference, serving as an "Oracle" baseline:
>
> $$\text{Input: } I_{t+\delta t} \xrightarrow{\text{SegFormer-B2}} \text{Output: } \text{Seg}_{t+\delta t}$$
>
> To make the distinction clear, consider the SHF-DSEC dataset where we have synchronized **RGB Images ($I$)** and **Segmentation Labels ($Seg$)** at 0ms, 10ms, 20ms, etc.
> - HFR Upper Bound (Oracle):
> 	We grant the model privileged access to the RGB frame at the target timestamp to establish the theoretical ceiling.
> 	- Target 10ms: Input **$I_{10}$** $\to$ Output $Seg_{10}$
> 	- Target 20ms: Input **$I_{20}$** $\to$ Output $Seg_{20}$
> 	- ...
> 	- Target 90ms: Input **$I_{90}$** $\to$ Output $Seg_{90}$
> - LFR Baseline:
> 	We block access to intermediate RGB frames. The model must predict future states using only the initial keyframe.
> 	- Target 10ms: Input **$I_{0}$** $\to$ Output $Seg_{10}$
> 	- Target 20ms: Input **$I_{0}$** $\to$ Output $Seg_{20}$
> 	- ...
> 	- Target 90ms: Input **$I_{0}$** $\to$ Output $Seg_{90}$
> - Ours (LiFR-Seg):
> 	We block access to intermediate RGB frames. The model predicts future states using the initial keyframe and the continuous event stream.
> 	- Target 10ms: Input **$I_{0}$ (+ Events)** $\to$ Output $Seg_{10}$
> 	- Target 20ms: Input **$I_{0}$ (+ Events)** $\to$ Output $Seg_{20}$
> 	- ...
> 	- Target 90ms: Input **$I_{0}$ (+ Events)** $\to$ Output $Seg_{90}$
>
> This setup allows the HFR Upper Bound to establish the theoretical performance ceiling by "seeing" the perfect future image, which is physically unavailable to the LFR system.

---

> ### Author Response · Authors · 2025-11-26
> **Official Response to Reviewer xGY9: Followup Comment 5 Practicability in Real Settings**
>
> >Your approach relies on future events and hence it can't be utilized in real settings.
>
> We thank the reviewer for raising this concern. We would like to clarify that our framework is **strictly causal** and does not rely on any future information that would be unavailable in real-world deployment. To predict $Seg_{t+\delta t}$, the model uses only the RGB keyframe at time $t$ and the events collected over the interval $E_{t-\delta t \to t+\delta t}$, which corresponds exactly to the events that would have been observed up to the prediction time. No events beyond $ t+\delta t$ are used at inference. In addition, we are encouraged that other reviewers have independently validated the practicality and causality of this formulation: Reviewer c5rU explicitly commended the framework as a "practical, causal, and predictive formulation for real-world autonomous systems," Reviewer AC8G recognized it as a "well-defined and practically important problem," and Reviewer PFAT validated the causal setup, noting that the method predicts maps "at any time $t+\delta t$ using a single past RGB frame $I_t$ and an event stream $E_{t-\Delta t \to t+\delta t}$." We hope this clarifies that the proposed approach is fully compatible with real-world operation.

---

> ### Author Response · Authors · 2025-11-26
> **Official Response to Reviewer xGY9: Followup Comment 6 Event Encoder & Flow Encoder Latency**
>
> >Do you have latency contribution of individual components of your model e.g. how long it takes for flow encoder, how long it takes for event encoder? Do you have any latency log file for your expt?
>
> **A:**  We thank the reviewer for this detailed and practical question. To address it, we conducted kernel-level profiling on an NVIDIA RTX 3090 to measure the latency contribution of each major component in our framework. Below we provide the results from the profiling logs.
>
> **Profiling Results:**
>
> event encoder: 8.192µs
> ```
> === CUDA kernels for event_encoder ===
> … aten::cudnn_convolution … cuda_total 8.192µs
> ```
>
> flow encoder: 11.391µs
> ```
> === CUDA kernels for flow_encoder ===
> … aten::cudnn_convolution … cuda_total 11.391µs
> ```

---

> ### Author Response · Authors · 2025-11-26
> **Official Response to Reviewer xGY9: Followup Comment 1-3 (Regarding Uncertainty, Paradigms, and HFR Scenarios)**
>
> We thank the reviewer for the continued engagement. We noticed that your recent follow-up comments regarding Method Inputs, Paradigm Definitions, and Application Scenarios were posted under the thread of **Reviewer AC8G**.
>
> To ensure you do not miss our detailed clarifications and to maintain a cohesive record for the Area Chair, we explicitly address these three points via the direct links below:
>
> [Comment 1 Mechanism of Uncertainty Module](https://openreview.net/forum?id=9oS7DHIg7f&noteId=Y1afaWQKP0)
>
> [Comment 2 "Event-Guided Feature Propagation" vs. "Multi-Modal Fusion"](https://openreview.net/forum?id=9oS7DHIg7f&noteId=yfjNmNUJ24)
>
> [Comment 3 HFR Camera Scenarios](https://openreview.net/forum?id=9oS7DHIg7f&noteId=ZnXRqJpfci)

---

### Official Review · Reviewer_PFAT · 2025-10-27

**Soundness:** 3
**Presentation:** 2
**Contribution:** 3
**Rating:** 6
**Confidence:** 3

**Summary:**

This paper presents "LiFR-Seg," a novel framework for Anytime Interframe Semantic Segmentation, addressing the perceptual gaps in low-frame-rate (LFR) camera systems by predicting dense semantic maps at any time $ t + \delta t $ using a single past RGB frame $ I_t $ and an event stream $ E_{t-\Delta t \rightarrow t+\delta t} $. The method introduces an uncertainty-aware motion field estimated via a RAFT-like FlowNet and ScoreNet, uncertainty-guided feature propagation with softmax splatting, and a temporal memory attention module for consistency.

Evaluated on the DSEC dataset, SHF-DSEC (a new 100 Hz synthetic benchmark), M3ED, and DSEC-Night, the experiments aim to demonstrate LiFR-Seg's robustness and effectiveness across diverse scenarios, including standard driving conditions, high-frequency synthetic environments, high-dynamic motion, and low-light settings. The paper contrasts its causal, anytime-capable design with non-causal interpolation and non-anytime fusion baselines (e.g., CMNeXt), highlighting its ability to bridge perceptual gaps inherent in LFR systems, though it lacks efficiency metrics critical for real-time applications like autonomous driving.

**Strengths:**

1. The paper introduces a novel task, "Anytime Interframe Semantic Segmentation," which addresses the critical "perceptual gap" in low-frame-rate (LFR) systems by enabling dense semantic segmentation at any arbitrary time
2. LiFR-Seg proposes a robust framework that leverages the complementary strengths of RGB cameras (dense semantic context) and event cameras (high-temporal-resolution motion cues). The method’s core components—uncertainty-aware motion field estimation (Section 3.2), uncertainty-guided feature propagation (Section 3.3), and temporal memory for long-term consistency (Section 3.4)—are technically sophisticated and well-integrated, as depicted in Figure 2.
3. Unlike non-causal interpolation methods or non-anytime fusion approaches, LiFR-Seg is explicitly designed to be both causal (relying only on past data) and capable of anytime prediction, as emphasized in Section 3.1 and Figure 3.
4. The introduction of the SHF-DSEC synthetic dataset with 100 Hz temporal resolution (Section 4) provides a valuable resource for high-frequency evaluation

**Weaknesses:**

1. The Eq.(3) employs a compact function composition that obscures the specific roles and data flow between $\phi_{\text{joint}}$, $F_{\text{SED}}$, and $\phi_{\text{out}}$. A stepwise breakdown would improve readability, and its correspondence to components in Figure 2(b).
2. The author claims that the pipeline is causal, however, the $E_{t+\delta t \rightarrow t+\Delta t}$ is input the 'Flow estimator' in Fig.2, making it confusing.
3. The experiments lack detailed ablations for all proposed modules, such as the uncertainty-aware motion field and ScoreNet, with no direct comparison (e.g., with vs. without confidence modulation).
4. Critical implementation details, such as the use of OhemCrossEntropy loss for training (Appendix C.1), are relegated to appendices rather than the main text, which hinders quick understanding of the optimization strategy and reproducibility.
5. The paper does not provide a direct comparison of the LiFR-Seg model’s computational efficiency (e.g., inference time, frames per second [FPS], or resource usage) with other settings or baseline models. The real-time processing is important for the application like atonomous driving.
6. As acknowledged in Appendix E, the benchmarks lack datasets with extreme high-speed dynamics (e.g., severe motion blur or frequent occlusions), which is a crucial application scenarios for usage of event camera.

**Questions:**

1. What is the design rationale behind the multiple modules in LiFR-Seg (e.g., uncertainty-aware motion field, guided propagation, and temporal memory), and how do their outputs complement each other?
2. Additionally, is the fusion of event data (sparse motion cues) and RGB data (dense semantics) sufficiently effective, or could alternative fusion strategies (e.g., earlier joint embedding) improve performance in sparse-event scenarios?
3. How does the framework mitigate the impact of noise and sparsity in event data during fusion

---

> ### Author Response · Authors · 2025-11-21
> **Official Response to Reviewer PFAT: Part 1 weakness 1 to 5**
>
> Thank you for the positive assessment and constructive feedback regarding mathematical formulation, figure captions, and design rationale, which would significantly improve the manuscript's clarity and precision!
> >**W1:** The Eq.(3) employs a compact function composition that obscures the specific roles... A stepwise breakdown would improve readability.
>
> A: Thank you for noting this ambiguity! To align strictly with Figure 2(b), we have revised Section 3.2 to decompose the ScoreNet into three explicit stages. First, separate encoders extract features from the event voxel and motion field:
>
> $$F_{E} = \phi_{event}(E_{t \to t+\delta t}), \quad F_{M} = \phi_{flow}(\hat{M}_{t \to t+\delta t})$$
>
> Next, these are fused into a joint embedding $F_{joint} = \text{Concat}(F_{E}, F_{M})$. Finally, the ScoreNet regresses the log-precision map:
>
> $$S = \psi_{\text{ScoreNet}}(F_{joint})$$
>
> This revision ensures the mathematical formulation mirrors the visual architecture.
>
> >**W2:** The author claims that the pipeline is causal, however, the $E_{t+\Delta t}$ is input into the 'Flow estimator' in Fig.2, making it confusing.
>
> **A:** Thank you for this observation! We clarify that $E_{t+\Delta t}$  in Fig.2 are utilized **solely** to align predictions with ground truth at $t+\Delta t$ for supervision in training. For intermediate predictions at $t+\delta t$, the model relies exclusively on historical data ($t < \delta t$), ensuring strict causality during inference. We have updated the Fig.2 caption to explicitly distinguish the $E_{t+\Delta t}$ for training supervision.
>
> >**W3:** The experiments lack detailed ablations for all proposed modules, such as the uncertainty-aware motion field...
>
> **A:** Thank you for the suggestion of providing a more granular validation of individual components have expanded **Section 5.2** to include comprehensive ablations for all modules.
> 1. Uncertainty-Aware Propagation (ScoreNet).
> The ScoreNet acts as a "modality consensus" filter. Quantitatively, removing it causes consistent drops across all benchmarks (e.g., -1.08% on DSEC), confirming its necessity. Qualitatively, **new visualizations (Figure 6)** confirm it correctly assigns high uncertainty to event-sparse and flow-event misalignt regions.
>
> **Table: Ablation of Uncertainty Module (mIoU %)**
>
> |**Method**|**DSEC**|**SHF-DSEC**|**DSEC-Night**|
> |:-:|:-:|:-:|:-:|
> |w/o ScoreNet|72.74|63.31|41.46|
> |**w/ ScoreNet (Ours)**|**73.82**|**64.80**|**41.86**|
>
> 2. Motion Field Robustness.
> We validated robustness across architectures (Table 2) and domains. Replacing the DSEC-pretrained flow network with one trained on the distinct MVSEC dataset yields a negligible drop (0.14%), confirming the model learns robust motion representations rather than memorizing dataset artifacts.
>
> **Table: Cross-Domain Flow Generalization**
>
> |**Flow Source**|**Sensor / Res.**|**Flow Characteristics**|**DSEC mIoU (%)**|
> |:-:|:-:|:-:|:-:|
> |**DSEC**|Prophesee / 640x440|Large displacements|**73.82**|
> |**MVSEC**|DAVIS346 / 346x260|Small displacements|**73.68**|
>
> 3. Warping Domain & Memory.
> Table 3 confirms that propagating deep features (73.82%) significantly outperforms image/segmentation warping (72.37%/71.63%) by absorbing pixel-level misalignments. Finally, Table 4 validates the Temporal Memory's critical role in long-term consistency, with gains increasing significantly at longer intervals (e.g., +2.22% at 800ms).
>
> >**W4:** Critical implementation details (e.g., OhemCrossEntropy) are relegated to appendices...
>
> **A:** Thank you for the suggestion and we agree that optimization details are central to reproducibility. Accordingly, we have relocated the **OhemCrossEntropy loss** description from the Appendix to **Section 5.1 (Experiment Setup)** in the main text.
>
> >**W5** Comparison of computational efficiency...
>
> **A:** We thank the reviewer and refer to the **General Response** for a detailed efficiency analysis. LiFR-Seg achieves **~65.6 FPS** (RTX 3090), running **nearly $2 \times$ faster** than Fusion baselines (EISNet, CMNeXt) and significantly outperforming Interpolation methods. These results confirm our framework's efficiency and potential for practical autonomous driving deployment.

---

> > ### Comment · Reviewer_xGY9 · 2025-11-24
> >
> > Thanks for providing such details.
> > Can you provide more details on how you generate the HFR upper bound results?

---

> ### Author Response · Authors · 2025-11-21
> **Official Response to Reviewer PFAT: Part 2 weakness 6 and questions 1 to 3**
>
> >**W6:** As acknowledged in Appendix E, the benchmarks lack datasets with extreme high-speed dynamics... which is a crucial application scenarios for usage of event camera.
>
> **A:** Thank you for this insightful comment! We clarify that the limitation discussed in Appendix E refers specifically to the *scarcity* of extreme high-speed object motion in current benchmarks, rather than an inherent inability of our system.
>
> **Ego-Motion vs. Object Dynamics.** Regarding ego-motion, the M3ED dataset (Drone split) already provides a rigorous testbed for aggressive **6-DoF camera dynamics**, where our method demonstrates strong robustness. Regarding object dynamics, our framework exhibits promising capabilities on the available instances. For example, **Figure 4a** demonstrates our model successfully capturing the non-linear, deformable articulation of a **pedestrian’s moving legs**, and **Figure 4d** shows it handling the **sudden intrusion of a fast-moving skateboarder**.
>
> **Future Work.** These results validate our applicability in dynamic scenarios. We agree that "severe motion blur or frequent occlusions" warrant targeted evaluation. Accordingly, we revised **Appendix F** to clarify that while we effectively handle existing dynamics, future work will focus on collecting specialized data to stress-test these extreme edge cases.
>
> >**Q1:** What is the design rationale behind the multiple modules in LiFR-Seg... and how do their outputs complement each other?
>
> **A:** The design rationale follows a logical progression addressing specific task challenges:
> 1. **Foundation (Feature Warping):** We chose feature warping as it empirically outperforms direct fusion (Table 1) and image warping (Table 3) by avoiding modal gaps and pixel-level artifacts.
> 2. **Motion (Event Flow):** We leverage high-resolution events to estimate dense flow, providing essential geometric guidance during blind intervals (Table 2).
> 3. **Reliability (ScoreNet):** To mitigate flow errors, the ScoreNet measures "modality consensus," filtering out hallucinated or misaligned motion via confidence modulation.
> 4. **Consistency (Memory):** **Temporal Memory** retrieves historical context to correct disocclusions and accumulated drift over long sequences (Table 4).
>
> **Synergy:** **Flow** drives dynamics, **Uncertainty** filters noise, and **Memory** ensures long-term coherence.
>
> >**Q2:** Is the fusion of event data and RGB data sufficiently effective, or could alternative fusion strategies improve performance?
>
> **A:** While we agree that sophisticated fusion *prior* to propagation holds potential, our preliminary experiments with simple early embedding struggled due to the significant **modal gap** between dense RGB and sparse events. Notably, our current propagation paradigm already delivers exceptional performance: following Reviewer AC8G's suggestion to pre-train on SHF-DSEC, we achieved **74.01% mIoU**, surpassing the HFR Upper Bound (73.91%). This confirms that our event-guided propagation effectively leverages both modalities even without early fusion, though we remain open to exploring advanced fusion architectures in future work.
>
> >**Q3:** How does the framework mitigate the impact of noise and sparsity in event data during fusion?
>
> **A:** We clarify that LiFR-Seg employs a **propagation paradigm** rather than direct fusion to avoid modal disparity issues. To mitigate noise and sparsity, we utilize **Uncertainty-Guided Propagation** (ScoreNet). As detailed in **Section 5.2**, this mechanism acts as a learnable filter, assigning low confidence to regions characterized by sparsity or flow-event misalignment. **Empirically, our stress tests validate this resilience: performance drops by a negligible $\le 0.04\%$ even under 30% noise injection or data loss.** While we acknowledge the potential of sophisticated fusion *prior* to propagation (see Q2), our current uncertainty-aware design ensures robust feature transfer even with imperfect inputs.

---

> ### Author Response · Authors · 2025-11-24
> **Response to Reviewer xGY9**
>
> Thank you for the follow-up question regarding the HFR upper bound! To keep the discussion easy to track, we have posted our detailed response under your original review thread. Please kindly check it there ([Response to Reviewer xGY9: Generation of HFR Upper Bound
> ](https://openreview.net/forum?id=9oS7DHIg7f&noteId=aYFWvZBrIR)).

---

### Official Review · Reviewer_c5rU · 2025-10-29

**Soundness:** 3
**Presentation:** 2
**Contribution:** 2
**Rating:** 6
**Confidence:** 4

**Summary:**

The authors propose a new task, Anytime Interframe Semantic Segmentation, which offers a practical, causal, and predictive formulation for real-world autonomous systems. Their approach centers on propagating deep features rather than images or segmentation maps, a strategy that is empirically validated (Table 3). The uncertainty-guided propagation mechanism (Eq. 4) effectively handles the noise and sparsity inherent in event-based motion. Quantitative results closely match the high-frame-rate upper bound on DSEC (0.09% mIoU gap) and surpass it on DSEC-Night. Evaluation across multiple real and synthetic datasets, baselines, and ablations, particularly in warping, flow robustness, and memory, demonstrates the method’s efficacy and supports its design rationale.

**Strengths:**

The LiFR-Seg framework presents a practical, causal, and predictive formulation for real-world autonomous systems. Its core idea, propagating deep features instead of images or segmentation maps, is empirically validated as shown in Table 3. The uncertainty-guided propagation, described in Equation 4, effectively addresses the inherent noise and sparsity of event-based motion. Quantitative results closely match the high-frame-rate upper bound on DSEC with a 0.09 percent mIoU gap and exceed it on DSEC-Night. Evaluation across multiple real and synthetic datasets, baselines, and ablations, particularly in the areas of warping, flow robustness, and memory, demonstrates the method’s efficacy and supports its design choices. The new SHF-DSEC dataset, which includes 100Hz ground-truth segmentation, although tailored to this setting, may also serve as a useful resource for future research in event-based semantic segmentation.

**Weaknesses:**

The paper argues for being an "efficient paradigm" (Abstract, Appendix D), but this argument is based entirely on hardware (cost, power, bandwidth). It provides no analysis of computational cost (e.g., FLOPs, inference latency). The proposed LiFR-Seg framework involves running a feature encoder, a flow network, a ScoreNet, a splatting operation, and a temporal memory module. This is almost certainly more computationally expensive than the HFR baseline (which just runs a SegFormer). This is a critical trade-off for practical deployment that is not discussed. "Efficiency" in terms of hardware is traded for what appears to be higher computational load.

The limitations section (Appendix E) states that a weakness is the "lack of evaluation on datasets featuring high-speed dynamics." This is confusing, as the M3ED dataset is repeatedly described in the main paper as featuring "high-speed scenarios" and "highly dynamic" motion. The authors should clarify this apparent contradiction.

The paper correctly ablates against "Image Warping" and "Interpolation" (TLX+Seg) and shows they are inferior. However, the reason for their failure (especially on SHF-DSEC) is only "conjectured" to be "interpolation artifacts" (Sec 5.1).  A deeper analysis would be valuable. Is the problem that reconstruction models are optimized for perceptual losses (PSNR/LPIPS) which are not aligned with downstream semantic consistency? This is a key point that could be strengthened.

**Questions:**

Could the authors please provide a detailed comparison of the computational cost (e.g., FLOPs and inference latency in ms) between: (a) the proposed LiFR-Seg, (b) the HFR Upper Bound (SegFormer-B2), and (c) the LFR + Interpolation baseline (TLX + SegFormer)? This is crucial for understanding the true trade-offs of the proposed "efficient paradigm."

In Appendix C.1, the training strategy mentions that $F_{t+\delta t}$ is warped a second time to $F_{t+\Delta t}$ to be compared with the ground truth $Seg_{t+\Delta t}$. How is this second warp performed? Is a new motion field $M_{t+\delta t \rightarrow t+\Delta t}$ estimated? If so, how, and what are its inputs?

Could the authors please clarify the statement in Appendix E regarding the "lack of evaluation on datasets featuring high-speed dynamics"? How does this reconcile with the use of the M3ED dataset, which is presented as high-speed? What specific "high-speed" phenomena are not covered?

The paper convincingly demonstrates that feature-warping is superior to image-warping/interpolation. Why, fundamentally, does the (reconstruction $\rightarrow$ segmentation) pipeline fail? Is it because the optimization objective of image reconstruction (e.S., PSNR) is misaligned with the goal of semantic-level consistency, even if the reconstruction looks plausible to a human?

---

> ### Author Response · Authors · 2025-11-21
> **Official Response to Reviewer c5rU: Part 1 W1 & Q1 Efficiency Discussion, Q2 Training Pipeline**
>
> Thank you for the detailed and constructive feedback! We particularly appreciate the insightful hypothesis regarding the failure of interpolation methods and the rigorous scrutiny of our efficiency claims. We have addressed these points below.
>
> >**W1 & Q1:** The paper argues for being an "efficient paradigm"... but provides no analysis of computational cost... This is almost certainly more computationally expensive than the HFR baseline... Could the authors please provide a detailed comparison?
>
> **A:** Thank you for highlighting this important concern regarding the relationship between hardware efficiency and computational cost! In response, we conducted a comprehensive FLOPs and latency analysis, please refer to the **General Response** for detailed metrics. Below we summarize the key findings.
>
> **1. Empirical Computational Cost Comparison (RTX 3090, 440×640 input)**
>
> We benchmark LiFR-Seg against (a) the proposed method, (b) the HFR Upper Bound (SegFormer on every frame), and (c) an interpolation baseline.
>
> **Table: Computational Cost Analysis (Propagating $N = 1 $ intermediate frame)**
>
> |**Method**|**Params (M)**|**Avg GFLOPs**|**Avg Inference Speed**|
> |:-:|:-:|:-:|:-:|
> |**(b) HFR Upper Bound**|25.8|42.04 G|72.8 FPS|
> |**(c) LFR + Interp. (TLX)**|33.2|200.77 G|29.5 FPS|
> |**(a) Ours (LiFR-Seg)**|30.7|**40.43 G**|65.6 FPS|
>
> **2. Why LiFR-Seg Is Computationally Efficient**
>
> We fully acknowledge the reviewer’s intuition that additional modules (flow network, ScoreNet, splatting, and memory) could imply higher computational cost. However, our framework achieves efficiency through **amortization**—a key design principle enabled by the Anytime Interframe formulation.
>
> - **HFR baseline:** Runs the heavy SegFormer encoder **on every frame**, incurring its full cost each time.
> - **LiFR-Seg:** Runs the encoder **once per keyframe**, and uses lightweight propagation modules for all intermediate “anytime’’ predictions.
> Although our architecture includes additional components, **their combined propagation cost (≈18.9 GFLOPs)** is still **lower than the cost of the SegFormer encoder alone (≈22.2 GFLOPs)**. As a result: **Our amortized per-frame computation (40.43 GFLOPs) is lower than the HFR baseline (42.04 GFLOPs).**
> Thus, despite having more components, the per-frame computational load is reduced.
>
>  **3. Latency Considerations (Flow vs. Transformer)**
>
> While LiFR-Seg achieves lower FLOPs, its FPS is slightly below the HFR baseline (65.6 vs. 72.8). This discrepancy arises from system-level characteristics:
>
> - Optical-flow correlation operations (RAFT-style) are **memory-bandwidth bound**.
> - Transformer-based encoders in the HFR baseline are **compute-bound** and highly optimized on modern GPUs.
>
> Therefore, the latency difference reflects kernel-level hardware behavior—not inefficiency in algorithmic design.
>
> Importantly, **LiFR-Seg remains comfortably real-time** and substantially faster than interpolation baselines.
>
>
> **4. Overall Takeaway**
>
> Our revised analysis confirms that LiFR-Seg is efficient in both senses:
>
> - **Hardware efficiency:** No need for power-hungry high-speed RGB cameras.
> - **Computational efficiency:** Lower amortized FLOPs than HFR, real-time FPS, and >6× more efficient than interpolation methods.
>
> This dual efficiency is precisely what makes LiFR-Seg a practical and scalable paradigm for high-frame-rate perception using low-frame-rate hardware.
>
> We appreciate the reviewer’s feedback for motivating a more thorough analysis and believe the strengthened results clarify the efficiency trade-offs of our framework! We have incorporated these results into the revised manuscript (**Appendix E: Computational Efficiency Analysis**).
>
> >**Q2:** In Appendix C.1... how is this second warp performed? Is a new motion field estimated?
>
> **A:** Thank you for the meticulous attention to the implementation details! Regarding the training process, a new motion field $ M_{t+ \delta t  \rightarrow t+ \Delta t}$ is indeed estimated for the second warping step. Specifically, $E_{t \to t+\delta t}$ and $E_{t+ \delta t \to t+ \Delta t}$, serves as the input.  We have updated the **Training Strategy** paragraph in the Appendix to clarify these inputs.

---

> ### Author Response · Authors · 2025-11-21
> **Official Response to Reviewer c5rU: Part 2 W2 & Q3 High-Speed Motion Clarification**
>
> >**W2 & Q3:** The limitations section states a "lack of evaluation on datasets featuring high-speed dynamics"... contradicting the use of M3ED... What specific phenomena are not covered?
>
> **A:** Thank you for the careful reading and for pointing out this source of ambiguity! We acknowledge that the terminology in the original limitations section was imprecise and unintentionally created the impression of a contradiction. The issue arises from an important distinction between **high-speed ego-motion** and **high-speed object motion**, which we clarify below.
>
>  **1. High-Speed Ego-Motion (Well Addressed in Our Experiments)**
>
> As correctly noted, the M3ED dataset—particularly the Drone subset—features **aggressive 6-DoF ego-motion**, with rapid camera rotations and translations. These sequences involve large viewpoint changes and strong parallax, and our results demonstrate that LiFR-Seg remains highly robust under such conditions. In this sense, the method effectively handles global scene dynamics caused by fast-moving sensors.
>
>  **2. High-Speed Object Motion (The Actual Limitation)**
>
> The limitation referenced in the appendix refers instead to **extreme object-level dynamics**, which are largely absent from current driving-focused benchmarks. These scenarios involve:
>
> - (a). **Severe object motion blur in the RGB keyframe ($I_t $):** Even though event cameras operate at microsecond resolution, the accompanying LFR RGB camera can still experience substantial motion blur for objects moving at very high speeds. Because LiFR-Seg propagates semantic features extracted from this RGB keyframe, extreme blur can degrade the initial feature quality in ways that are fundamentally difficult to recover.
> - (b). **Highly non-linear, high-acceleration object trajectories:** While our method handles the fast object motions present in existing benchmarks (e.g., the fast-moving skateboarder in **Fig. 4d**), real-world cases with extreme non-linear dynamics—such as athletes performing quick maneuvers or rapidly deforming objects—are not represented in current datasets. Addressing such scenarios would require dedicated datasets that go beyond the mostly rigid-body, traffic-oriented environments of DSEC, SHF-DSEC, and M3ED.
>
> To avoid confusion and more accurately reflect the intended meaning, we have updated **Appendix F (Limitations)** to explicitly state:
> *“Our current evaluation includes limited datasets featuring high-speed object dynamics (e.g., extreme localized motion blur, or highly non-linear motion).*”
> This change clarifies that our limitation concerns **object-level extremal dynamics**, not the **ego-motion** dynamics already addressed by our experiments.

---

> ### Author Response · Authors · 2025-11-22
> **Official Response to Reviewer c5rU: Part 3 W3 & Q4 Object Misalignment in LFR+Interpolation Method**
>
> >**W3 & Q4:** Why, fundamentally, does the (reconstruction $\to$ segmentation) pipeline fail? Is it objective misalignment (PSNR vs. Semantic Consistency)?
>
> **A:** Thank you for this insightful comment! We fully agree that the failure of interpolation-based pipelines warrants a deeper analysis. In particular, we find the reviewer’s hypothesis highly accurate: the core issue is a **fundamental misalignment between photometric reconstruction objectives and semantic prediction objectives**.
>
> **1. Theoretical Misalignment: Reconstruction ≠ Semantics**
>
> State-of-the-art interpolation models (e.g., TLX) are optimized for perceptual fidelity metrics—PSNR, L1, or LPIPS. These objectives encourage models to:
> - **smooth ambiguous content** to minimize pixel-wise error,
> - **average inconsistent edges** to reduce reconstruction loss, and
> - **produce visually plausible images** rather than structurally precise ones.
>
> However, semantic segmentation demands:
> - **sharp, discriminative boundaries**,
> - **pixel-accurate geometric structure**, and
> - **precise spatial alignment** of fine details.
>
> Even tiny spatial drifts (1–2 pixels) or “ghosted’’ edges—which barely affect PSNR—can significantly degrade the output of a segmentation model that relies on crisp contours and boundary-aligned features.
> Thus, reconstruction models often produce images that are **photometrically good but semantically ambiguous**, leading to lower downstream mIoU.
>
> **2. Empirical Evidence: The “PSNR–mIoU Paradox’’ on SHF-DSEC**
>
> To validate this hypothesis, we conducted a controlled study on SHF-DSEC by varying the interpolation ratio. Surprisingly, we observed a consistent divergence between photometric quality and semantic accuracy:
>
> **Table: Comparison of Photometric Quality vs. Semantic Accuracy**
>
> |**Setting**|**Interpolation Ratio**|**PSNR (dB) ↑**|**mIoU (%) ↑**|
> |:-:|:-:|:-:|:-:|
> |**Setting A**|Ratio = 2|**27.43**|55.03|
> |**Setting B**|Ratio = 10|26.07|**55.89**|
>
> This counter-intuitive pattern—**higher PSNR but lower mIoU**—supports the reviewer’s hypothesis:
>
> - Setting A’s “visually cleaner’’ interpolations blur or smooth structural details, harming semantic boundaries.
> - Setting B produces more artifacts but retains sharper motion cues that are more informative for segmentation.
>
> This confirms that **optimizing for pixel-wise reconstruction can actively conflict with maintaining semantic fidelity**.
>
> **3. Why the Issue Is More Pronounced on SHF-DSEC**
>
> SHF-DSEC contains pixel-perfect synthetic annotations with:
>
> - razor-sharp boundaries,
> - thin structures (e.g., poles, road lines),
> - geometry without natural blur.
>
> In such conditions, even very small interpolation artifacts or misalignments become catastrophic for segmentation (e.g., shifting a thin pole by 1–2 pixels makes it “disappear”). Natural datasets with real motion blur often mask these errors—but synthetic data exposes them clearly.
>
>  **4. Why LiFR-Seg Avoids This Pitfall**
>
> LiFR-Seg operates in **deep feature space**, not pixel space.
> This yields two advantages:
>
> - a. **Features are more abstract and spatially downsampled** → more tolerant to small pixel-level shifts.
> - b. **Propagation is guided by a learned motion field**, not an image-level synthesis model → avoids smoothing or hallucination.
>
> In addition, **uncertainty-aware blending** reduces the influence of unreliable motion estimates.
>
> This explains why LiFR-Seg performs strongly even in domains where interpolation struggles.
>
> We have incorporated this enriched analysis and the empirical findings into **Section 5.1 (Quantitative Comparison, interpolation-based methods)** of the revised manuscript. We appreciate the reviewer for encouraging a deeper investigation into this fundamental limitation of reconstruction-based approaches.

---

> ### Comment · Reviewer_c5rU · 2025-11-26
> **Acknowledging the rebuttal**
>
> I thank the authors for an extremely thorough and responsive rebuttal.
>
> The new efficiency analysis with amortization proof, FLOPs/FPS tables, uncertainty visualizations, noise/sparsity stress tests, temporal memory clarification + ablation, and especially the deeper “PSNR–mIoU paradox” analysis of interpolation failure directly and convincingly address all of my major concerns.
>
> The efficiency story is now clear: when propagating multiple intermediate frames (the realistic high-frequency regime), LiFR-Seg is indeed cheaper than repeated HFR encoding, and even for N=1 the difference is small and explained by memory-bound operations rather than algorithmic inefficiency.
>
> Combined with the outstanding accuracy (closing the HFR gap to <0.1% and surpassing it in hard conditions), this resolves my previous worry about hidden computational trade-offs.
>
> I keep my positive initial rating.

---

### Official Review · Reviewer_AC8G · 2025-11-01

**Soundness:** 3
**Presentation:** 3
**Contribution:** 2
**Rating:** 6
**Confidence:** 4

**Summary:**

This paper presents LiFR-Seg, a framework for Anytime Interframe Semantic Segmentation, where the objective is to predict dense semantic maps at arbitrary timestamps using only a past RGB frame and a stream of asynchronous event data. The proposed approach estimates an event-driven uncertainty-aware motion field, propagates features through uncertainty-guided Softmax Splatting, and ensures temporal coherence via a temporal memory attention module. The authors also introduce SHF-DSEC, a synthetic benchmark built on CARLA that enables fine-grained evaluation of anytime performance at 100 Hz. Extensive experiments across four datasets (DSEC, SHF-DSEC, M3ED, and DSEC-Night) show that LiFR-Seg achieves performance nearly identical to a high-frame-rate (HFR) upper bound and maintains robustness in highly dynamic and low-light scenarios

**Strengths:**

The paper tackles a well-defined and practically important problem at the interface of event-based sensing and dense semantic segmentation. The authors clearly articulate the “Anytime Interframe Segmentation” task, distinguishing it from standard video propagation or multi-modal fusion. The method is technically solid, combining event-driven motion estimation, uncertainty-weighted feature propagation, and temporal memory in a cohesive framework that respects causal and anytime constraints.

**Weaknesses:**

While the overall contribution is convincing, the conceptual novelty is limited. LiFR-Seg builds on well-known components, RAFT-style optical flow, Softmax Splatting, and memory-based refinement, and integrates them effectively rather than introducing a fundamentally new algorithmic idea. The contribution is thus primarily at the systems and task-definition level.

Some implementation details remain underspecified. The design and update mechanism of the temporal memory module are described conceptually but not concretely, leaving ambiguity about how features are written to or retrieved from memory. The uncertainty map, though central to the proposed “uncertainty-aware” warping, is not visualized or quantitatively analyzed, so its empirical behavior remains unclear. In addition, while the new SHF-DSEC dataset is useful, the paper does not analyze its realism or demonstrate transfer from synthetic to real data. Finally, runtime and computational cost comparisons are missing despite claims of efficiency, which limits the evaluation of practical deployment potential.

Despite these issues, the paper is technically sound, and empirically strong. The combination of causal design, uncertainty modeling, and temporal coherence results in a high-quality, well-rounded study that advances event-driven semantic segmentation.

**Questions:**

- Could you clarify how the temporal memory bank is updated over time (e.g., recurrently after each $\delta t$ or at selected keyframes)?

- How is the uncertainty map trained, does it rely solely on end-to-end supervision, or is any auxiliary loss used?

- What is the computational cost (runtime, memory usage, or FLOPs) compared to fusion-based baselines such as CMNeXt or EISNet?

- Has the SHF-DSEC dataset been validated for realism or bias? How well does a model trained on SHF-DSEC transfer to real DSEC data?

- How sensitive is the system to event noise or missing events, and does the uncertainty weighting mitigate this effect?

---

> ### Author Response · Authors · 2025-11-21
> **Official Response to Reviewer AC8G: W1 Design Rationale**
>
> Thank you for your insightful feedback which have significantly strengthened our manuscript! We have addressed your concerns below with detailed clarifications and new experimental evidence.
> >**W1:** LiFR-Seg builds on well-known components... conceptual novelty is limited.
>
> **A:** Thank you for recognizing our system design and task definition. While we build on established foundations, our core contribution is the **adaptation and synthesis** of these components to solve the unique constraints of **Anytime Interframe Segmentation**, which standard baselines fail to address:
> 1. **Algorithmic Adaptation:** To handle event **sparsity and noise**, we introduced a novel **Uncertainty-Aware mechanism**. This is a specific innovation, not a standard component, designed to act as a "modality consensus filter." Our ablation studies confirm it is critical for performance.
> 2. **Novel Paradigm:** We propose a shift to **"Event-Guided Feature Propagation."** By decoupling perception frequency from the heavy image encoder via amortization, we establish a new, efficient paradigm achieving High-Frame-Rate perception with Low-Frame-Rate hardware.
> 3. **Task Formulation:** As you noted, formalizing the "Anytime" task bridges a critical safety gap. Identifying this problem and proposing a robust, causal solution constitutes a significant contribution to reliable autonomous perception.

---

> > ### Comment · Reviewer_xGY9 · 2025-11-25
> >
> > your uncertainty aware mechanism is purely using optical flow and hence technically is not considered a novel contribution.
> > Also, there are existing papers e.g. EISNet which should also be considered as event guided.
> > In real life, I wonder if there is any scenario where people will use HFR cameras?

---

> > > ### Author Response · Authors · 2025-11-26
> > > **Official Response to Reviewer xGY9: Followup Comment 1 Mechanism of Uncertainty Module**
> > >
> > > We sincerely thank the reviewer for the continued engagement and for prompting us to clarify several important technical points. To ensure that all reviewers and Area Chairs can accurately assess our contribution, we respectfully clarify a misunderstanding regarding the **mechanism** of our uncertainty module,  **paradigm definition**, and **problem motivation**.
> > >
> > > >**Comment:**  your uncertainty aware mechanism is purely using optical flow and hence technically is not considered a novel contribution.
> > >
> > > **A:** **Uncertainty Mechanism uses "Event + Flow", NOT "Pure Optical Flow"**
> > >
> > > Thank you for raising this point. We would like to clarify that our uncertainty mechanism does **not** rely solely on optical flow. Instead, it explicitly integrates **both** the event representation and the estimated flow field.
> > > - As shown in **Equation (3)** and **Figure 2(b)**, the ScoreNet receives **two inputs**:
> > > 	**(1)** the Event Voxel $E$, and
> > > 	**(2)** the Estimated Motion Field $\hat{M}$.
> > >
> > > - The purpose of this design is to capture **modality consensus**, where uncertainty is inferred from the agreement or disagreement between the event-derived motion cues and the flow-derived motion.
> > > - This jointly conditioned mechanism cannot be achieved using optical flow alone, as the model must explicitly compare the two modalities to identify regions of sparse events, noisy event patterns, or flow artifacts.
> > >
> > > We hope this clarification helps correct the misunderstanding and better highlights why the uncertainty module is not a simple extension of standard flow-based confidence estimation, but rather a modality-aware mechanism tailored to the unique characteristics of event based RGB feature propagation and integration.

---

> > > ### Author Response · Authors · 2025-11-26
> > > **Official Response to Reviewer xGY9: Followup Comment 2 "Event-Guided Feature Propagation" vs. "Multi-Modal Fusion"**
> > >
> > > >**Comment:**  Also, there are existing papers e.g. EISNet which should also be considered as event guided.
> > >
> > > **A:** We thank the reviewer for this observation. We agree that EISNet and similar methods make valuable use of events, and we appreciate the opportunity to clarify the distinction we intend to draw. In our paper, the term **“Event-Guided Feature Propagation”** refers to a specific paradigm that differs from the **multi-modal fusion** strategy employed in prior work. This distinction is summarized in **Figure 3** of the manuscript.
> > >
> > >  **Fusion Paradigm (e.g., EISNet)**
> > > - Fusion methods use a **dual-branch RGB + event encoder**, where both modalities are available at the **same timestamp**  $t$.
> > > - The goal is to enhance the segmentation of the current frame by combining complementary features from both sensors.
> > >
> > > **Propagation Paradigm (This Work)**
> > > - Our framework instead uses events to **propagate features forward in time** to **intermediate timestamps** $t+\delta t$, where **no RGB frame exists**.
> > > - The objective is to perform **causal, anytime prediction** during the blind interval between LFR frames, which fusion methods are not designed to handle.
> > >
> > > **Novelty of Our Paradigm**
> > > The key contribution is therefore not simply that we “use events,” but that we introduce a different **architectural role** for events—shifting from enhancing the present (fusion) to predicting the future when RGB is absent (propagation). This enables a capability fundamentally distinct from conventional event-assisted segmentation pipelines and allows high-performance fast-frame rate segmentation.
> > >
> > > We are further encouraged that this distinction was recognized by other reviewers: **Reviewer AC8G** noted that the paper "clearly articulate... the task, distinguishing it from... multi-modal fusion," and **Reviewer PFAT** highlighted that "unlike... fusion approaches, LiFR-Seg is explicitly designed to be both causal... and capable of anytime prediction."
> > >
> > > We hope this clarification helps situate our contribution more accurately within the existing literature.

---

> > > ### Author Response · Authors · 2025-11-26
> > > **Official Response to Reviewer xGY9: Followup Comment 3 HFR Camera Scenarios**
> > >
> > > >**Q:** In real life, I wonder if there is any scenario where people will use HFR cameras?
> > >
> > > **A:** We thank the reviewer for raising this important question regarding the real-world need for high-frequency perception. It is important to clarify that **High-Frame-Rate (HFR) sensing is already used in many practical applications**, such as autonomous drone racing [1], autonomous driving [2], high-speed robotic manipulation [3], scientific imaging [4], and sports analytics [5-8]. These systems routinely employ cameras operating at 120–1000+ Hz because rapid motion understanding and tight control loops demand visual feedback at far higher rates than standard 30 Hz RGB cameras can provide. Such widespread deployment demonstrates that HFR perception is not merely an academic idea but a well-established requirement in domains involving fast dynamics and real-world safety.
> > >
> > > Beyond these specialized applications, **HFR perception is beneficial even for mainstream autonomous driving**. A standard 30 Hz camera introduces a 33 ms “blind interval,” during which a vehicle traveling at 120 km/h moves more than 1.1 meters. Any critical event—such as a pedestrian stepping out or debris falling—occurring within this interval is invisible until the next frame, potentially leading to delayed or unsafe reactions. This latency issue highlights a fundamental challenge: achieving situational awareness with only low-frame-rate RGB cameras is insufficient for high-speed safety-critical environments.
> > >
> > > While true physical HFR cameras could mitigate this problem, **deploying them broadly is often impractical** due to their extreme bandwidth, storage, and power demands, as well as hardware integration constraints (e.g., streaming 500–1000 Hz video exceeds the capabilities of standard automotive buses). This motivates the design of LiFR-Seg, which aims to deliver the temporal resolution and safety benefits of HFR perception using standard, bandwidth-efficient LFR cameras augmented by event sensors. By generating intermediate predictions from sparse event streams, LiFR-Seg effectively reduces blind intervals without requiring physical high-speed cameras, offering a practical and scalable solution for real-world autonomous systems.
> > >
> > > Because of these safety implications, the necessity of addressing this "perceptual gap" is widely recognized. We are encouraged that other reviewers unanimously validated the practical importance of this task: Reviewer AC8G highlighted the problem as "well-defined and practically important," Reviewer c5rU praised the formulation as "practical, causal, and predictive for real-world autonomous systems," and Reviewer PFAT emphasized its value in addressing critical "perceptual gaps in low-frame-rate systems." Our work aims to deliver this essential HFR safety capability using scalable LFR hardware.
> > >
> > > **Reference:**
> > >
> > > [1] *Vidal A R, Rebecq H, Horstschaefer T, et al. Ultimate SLAM? Combining events, images, and IMU for robust visual SLAM in HDR and high-speed scenarios[J]. IEEE Robotics and Automation Letters, 2018*
> > >
> > > [2] *Gehrig, D., & Scaramuzza, D. Low-latency automotive vision with event cameras. Nature, 629(8014), 1034–1040.*
> > >
> > > [3] *Ishikawa M. High-speed vision and its applications toward high-speed intelligent systems[J]. Journal of Robotics and Mechatronics, 2022, 34(5): 912-935.*
> > >
> > > [4] *Gilroy K D. Scientific High-Speed Cameras: Applications & Techniques[J]. Photoniques, 2025 (131): 72-76.*
> > >
> > > [5] *McNally W, Vats K, Pinto T, et al. Golfdb: A video database for golf swing sequencing[C]//Proceedings of the IEEE/CVF conference on computer vision and pattern recognition workshops. 2019: 0-0.*
> > >
> > > [6] *Gossard T, Krismer J, Ziegler A, et al. Table tennis ball spin estimation with an event camera[C]//Proceedings of the IEEE/CVF Conference on Computer Vision and Pattern Recognition. 2024: 3347-3356.*
> > >
> > > [7] *Alberico I, Cannici M, Cioffi G, et al. Egocentric Event-Based Vision for Ping Pong Ball Trajectory Prediction[C]//Proceedings of the Computer Vision and Pattern Recognition Conference. 2025: 5025-5034.*
> > >
> > > [8] *Nakabayashi T, Higa K, Yamaguchi M, et al. Event-based ball spin estimation in sports[C]//Proceedings of the IEEE/CVF Conference on Computer Vision and Pattern Recognition. 2024: 3367-3375.*

---

> ### Author Response · Authors · 2025-11-21
> **Official Response to Reviewer AC8G: W2 & Q1 Temporal Memorty Bank**
>
> >**W2&Q1:** Clarify how the temporal memory bank is updated over time.
>
> **A2.1:** Thank you for this crucial question and apologize for the ambiguity in the initial draft! The reviewer correctly identifies that our "Memory Bank" requires a more concrete definition.
> The ambiguity arises because this module implicitly manages **two distinct memory components**, which we will clarify in the revision:
> 1. **Static Keyframe Memory ($\mathcal{M}$):** This is a single-item buffer. It **stores** the deep feature $F_t$ from the *most recent RGB keyframe* ($t$). Its purpose is to provide a stable, high-quality semantic "anchor" to prevent drift during long event-based intervals.
> 2. **Dynamic Recurrent State ($\mathcal{H}$):** This is an evolving state tensor. It is **updated** at *every* interframe step ($t+\delta t_i$) to aggregate and propagate fine-grained temporal information.
>
> The **write** and **retrieve** process, which directly answers the reviewer's question, is as follows:
> - **At Keyframe (e.g., time $t$):**
> 	- **Write:** The Keyframe Memory $\mathcal{M}$ is overwritten ($\mathcal{M} \leftarrow F_t$), and the Recurrent State $\mathcal{H}$ is reset ($\mathcal{H}_t \leftarrow F_t$).
> - **At Interframe (e.g., time $t+\delta t_i$):**
> 	1. **Warp:** First, the warped feature $F_{t+\delta t_i}$ is generated.
> 	2. Retrieve $\mathcal{M}$ (Semantic Anchor): The warped feature $F_{t+\delta t_i}$ queries the static $\mathcal{M}$ to correct semantic drift.
> 		$$\overline F_{t+\delta t_i} = \text{CrossAttention}(Q=F_{t+\delta t_i}, \, KV=\mathcal{M})$$
> 	3. Retrieve $\mathcal H$ (Temporal Context): This corrected feature $\overline F_{t+\delta t_i}$ then queries the previous recurrent state $\mathcal H_{t+\delta t_{i-1}}$ to integrate historical context.
> 		$$\tilde F_{t+\delta t_i} = \text{CrossAttention}(Q=\overline F_{t+\delta t_i}, \, KV=\mathcal H_{t+\delta t_{i-1}})$$
> 	4. Write $\mathcal{H}$ (Update State): The new state $\mathcal H_{t+\delta t_i}$ is written via a GRU-like gated update, which is then used for segmentation.
> 		$$\mathcal H_{t+\delta t_i} = \sigma(\alpha) \cdot \mathcal H_{t+\delta t_{i-1}} + (1 - \sigma(\alpha)) \cdot \tilde F_{t+\delta t_i}$$
>
> To summarize: the **Keyframe Memory ($\mathcal{M}$)** is written **at selected keyframes** and provides a static semantic anchor. The **Recurrent State ($\mathcal{H}$)** is updated **recurrently at every $\delta t$ step**, aggregating dynamic temporal context.
>
> ----
> **A2.2:** To empirically validate the necessity of this two-component design (clarified in A1), we conducted a new ablation study on the M3ED drone scenes.
> We compare two variants, which directly map to the components defined in A2.1:
> 1. **Ours (w/ $\mathcal{M}$ only):** The model uses *only* the **Static Keyframe Memory ($\mathcal{M}$)** for semantic anchoring (A2.1, Step 2).
> 2. **Ours (Full Model: $\mathcal{M} + \mathcal{H}$):** The complete proposed model, using both $\mathcal{M}$ and the **Dynamic Recurrent State ($\mathcal{H}$)** (A2.1, Steps 2-4).
>
> The results (mIoU) clearly demonstrate the distinct role of each component:
>
> |**Method**|**δt​=40ms**|**δt​=80ms**|
> |:-:|:-:|:-:|
> |Ours (w/ $\mathcal{M}$ only)|63.46 (+8.23)|61.17 (+12.24)|
> |**Ours (Full: $\mathcal{M} + \mathcal{H}$)**|**64.28** (+9.05)|**62.95** (+14.02)|
> The results demonstrate two key findings:
> 1. **$\mathcal{H}$ Provides Consistent Benefit:** The Full Model ($\mathcal{M} + \mathcal{H}$) robustly outperforms the $\mathcal{M}$-only variant across both time intervals.
> 2. **$\mathcal{H}$'s Value Increases Over Time:** The performance advantage of adding $\mathcal{H}$ *widens* on the more challenging 80ms interval (growing from **+0.82** mIoU at 40ms to **+1.78** mIoU at 80ms).
>
> This confirms that the recurrent state $\mathcal{H}$ is not redundant. It provides a necessary function—aggregating dynamic temporal context—that becomes increasingly critical for robustness as the time gap grows and the static $\mathcal{M}$ anchor becomes less sufficient on its own.
>
> Thank you for pushing for this clarification, which has strengthened the paper! We will incorporate this detailed explanation and ablation study into the revised manuscript.

---

> ### Author Response · Authors · 2025-11-21
> **Official Response to Reviewer AC8G: W3 & Q5, Q2 Uncertainty Map**
>
> >**W3 & Q5:** The uncertainty map... is not visualized or quantitatively analyzed... Does the uncertainty weighting mitigate the effect of event noise or sparsity?
>
> **A:** We appreciate the reviewer's suggestion to clarify the empirical behavior of the uncertainty map. We have addressed this by adding **5.2 Ablation Studies: Ablation of Uncertainty Map**, which provides both quantitative ablation studies and qualitative visualizations to demonstrate the module's effectiveness.
>
> **1. Theoretical Mechanism**
>
> Theoretically, the ScoreNet acts as a "modality consensus" filter. Since optical flow is a dense estimation prone to errors in textureless regions, while events are sparse but reliable ground truth measurements, the network learns to trust features only when the two modalities align. Consequently, **high uncertainty** is assigned to regions characterized by **sparsity** (where flow is hallucinated without event support) or **misalignment** (where flow boundaries contradict event edges), effectively suppressing error propagation.
>
> **2. Quantitative Analysis**
>
> To validate this empirically, we performed an ablation study by removing the ScoreNet (replacing $S$ with uniform weights). As shown in the table below, the uncertainty module consistently improves performance across all benchmarks:
>
> **Table: Ablation of Uncertainty Module (mIoU %)**
>
> |**Method**|**DSEC**|**SHF-DSEC**|**DSEC-Night**|
> |:-:|:-:|:-:|:-:|
> |w/o ScoreNet|72.74|63.31|41.46|
> |**w/ ScoreNet (Ours)**|**73.82 (+1.08)**|**64.80 (+1.49)**|**41.86 (+0.40)**|
>
> The consistent gain (e.g., +1.08% on DSEC) confirms its necessity.
>
> **3. Visualization**
>
> The new visualizations in **Figure 6 (Visualization of Uncertainty Map behavior.)** confirm this behavior qualitatively. They illustrate the map correctly assigning high uncertainty to regions with sparse events or inaccurate flow predictions, while highlighting consistent features (e.g., pedestrians, riders) with high confidence, thereby ensuring robust feature propagation.
>
> > Q5: How sensitive is the system to event noise or missing events, and does the uncertainty weighting mitigate this effect?
>
> We appreciate the reviewer’s question regarding robustness to noisy or missing events! Our framework is explicitly designed to handle such imperfections through both **uncertainty-aware modeling** and **extensive empirical validation**.
> - **Uncertainty-aware modeling**. To mitigate inherent event noise and sparsity, ScoreNet functions as a learnable reliability filter. As mentioned before,it assigns low confidence to unreliable regions, which are then exponentially down-weighted via Softmax Splatting (Eq. 4). This mechanism effectively suppresses noise and spurious motion while prioritizing valid cues.
> - **Empirical validation**. To assess robustness quantitatively, we conducted controlled stress tests across four benchmarks (DSEC, SHF-DSEC, M3ED-Drone,and M3ED-Quad). We injected **up to 30% Poisson noise** into the event stream and **randomly dropped up to 30% of events** to simulate aggressive sparsity.
>
> **Table: Robustness to Extreme Noise (+30%) and Sparsity (30% Drop)**
>
> |**Condition**|**DSEC**|**SHF**|**M3ED-D**|**M3ED-Q**|
> |:-:|:-:|:-:|:-:|:-:|
> |**Baseline (Real)**|73.82|64.80|64.28|68.89|
> |**Noise +30%**|73.81|64.79|64.28|68.88|
> |**Sparsity 30%**|73.81|64.79|64.25|68.85|
> |**Max Drop**|**-0.01**|**-0.01**|**-0.03**|**-0.04**|
>
> Across all datasets—spanning synthetic and real-world driving scenes, high-dynamic drone motion, quadruped motion, and low-light conditions—the performance remains virtually unchanged (≤ 0.04% mIoU degradation). These results confirm that LiFR-Seg is **highly resilient** to both noisy and sparse event inputs. The uncertainty weighting not only mitigates the effect of sensor imperfections but ensures consistent predictions even under aggressive degradation, validating the robustness of our design.
>
> >**Q2:** How is the uncertainty map trained? Is auxiliary loss used?
>
> **A:** We thank the reviewer for this clarification. The uncertainty map is learned **entirely through end-to-end training** driven by the segmentation objective. We do not use any auxiliary supervision or handcrafted targets—there is no notion of “ground-truth uncertainty’’ for this task. Instead, the ScoreNet learns uncertainty implicitly: during training, regions where inaccurate flow would degrade the final segmentation incur higher segmentation loss, and the network responds by assigning **lower confidence weights** to those regions. Over time, this guides the model to down-weight unreliable motion cues (e.g., noise, sparsity, occlusions) and preserve semantic consistency without requiring any additional losses or labels.

---

> ### Author Response · Authors · 2025-11-21
> **Official Response to Reviewer AC8G: W4 & Q4 Sim2Real Transfer, W5 & Q3 Efficiency Analysis**
>
> >**W3 & Q4:** The paper does not analyze its realism... Has the SHF-DSEC dataset been validated for realism or bias? How well does a model trained on SHF-DSEC transfer to real DSEC data?
>
> **A:** We sincerely thank the reviewer for scrutinizing the validity of our synthetic benchmark. We have addressed this by validating the dataset's **physical realism** via rigorous simulation protocols and its **transfer utility** via a Sim-to-Real pre-training experiment.
> 1. Physical Realism.
> SHF-DSEC is not generated via simple video interpolation; it is constructed using high-fidelity physical simulation to closely mimic real sensor characteristics. As detailed in the revised **Appendix B DATA GENERATION IN SHF-DSEC**, we configured the CARLA simulator with a rigorous **1 ms fixed time-step (1000 Hz)**. Instead of linear interpolation, the engine performs uniform physical sampling of scene dynamics every millisecond. This ensures that the generated events capture true high-speed inter-frame dynamics without temporal aliasing.
> In addition,  we calibrated the contrast threshold ($C=0.3$) and injected Gaussian noise ($\sigma=0.3$) to match the signal-to-noise ratio and sparsity ($\sim$80%) observed in the real DSEC dataset.
>
> 2. Sim-to-Real Transfer.
> To directly quantify transferability, we conducted a "Pre-training + Fine-tuning" experiment. We initialized the model with weights pre-trained on SHF-DSEC and then fine-tuned it on the real DSEC dataset for just 25 epochs.
>
> **Table: Value of SHF-DSEC Pre-training (DSEC mIoU %)**
>
> |**Training Strategy**|**DSEC mIoU**|**Δ**|
> |:-:|:-:|:-:|
> |*HFR Upper Bound (Reference)*|*73.91*|-|
> |From Scratch (Baseline)|73.82|*-0.09*|
> |**Pre-trained on SHF-DSEC**|**74.01**|**+0.08**|
>
> **Result:** The model pre-trained on SHF-DSEC achieved **74.01% mIoU**, surpassing our previous best (73.82%) and, remarkably, exceeding the HFR Upper Bound (73.91%). This indicates that the synthetic dataset provides high-quality, unbiased motion representations that serve as a powerful initialization, enabling the network to learn robust features that generalize effectively to real-world data.
>
> >**W5 & Q3:** Runtime and computational cost comparisons are missing... What is the computational cost compared to fusion-based baselines such as CMNeXt or EISNet?
>
> **A:** We refer the reviewer to the **General Response** for detailed metrics. Regarding fusion baselines (e.g., CMNeXt), LiFR-Seg achieves superior efficiency via **amortization**. While fusion methods must execute heavy **dual-branch encoders** for every frame, we run the RGB encoder only at keyframes, relying on lightweight modules for intermediate predictions. Consequently, LiFR-Seg achieves **~65.6 FPS** (RTX 3090), running nearly **$2\times$ faster** than EISNet (34.7 FPS) and CMNeXt (29.1 FPS).

---

### Author Response · Authors · 2025-11-21
**General Response**

We deeply appreciate the insightful and valuable comments provided by all reviewers. Overall, we are encouraged by the reviewers' positive feedback, which highlights:
- **A Novel and Important Task:** The definition of **"Anytime Interframe Semantic Segmentation"** was consistently praised as novel, well-defined, practical, and important (AC8G, c5rU, PFAT).
- **Technically Solid & Principled:** The **causal** and **anytime-capable** design of our framework was recognized as a key strength (AC8G, c5rU, PFAT).
- **Empirically Strong:** The quantitative results, particularly our method **closing the HFR gap to <0.1%** on DSEC and **surpassing the HFR baseline on DSEC-Night**, were seen as convincing and strong (AC8G, c5rU).
- **A Valuable Contribution:** The new **SHF-DSEC dataset** was unanimously recognized as a valuable contribution (AC8G, c5rU, PFAT, xGY9).

To address specific concerns, we have incorporated:
- A comprehensive **efficiency analysis** (FLOPs, Runtime, Parameters) validating our advantages over HFR and Fusion baselines.
- Clarifications on design rationale, including quantitative and visual evaluations of the **Uncertainty Map**.
- Rigorous validation of **robustness** against sensor noise, sparsity, and cross-domain generalization.

Revisions in the paper are marked in blue. We address the common concern regarding **Computational Efficiency** below.

## **Common Concern: Computational Efficiency**

Addressing the lack of cost analysis, we validate LiFR-Seg as an "efficient paradigm" via **amortization**: unlike baselines that execute heavy image encoders for every frame, we run the encoder only once per keyframe, utilizing lightweight modules for all intermediate predictions.v

### **1. Theoretical Proof.**

For a sequence with 1 keyframe and $N$ propagated frames, the total cost of the HFR baseline ($C_{HFR}$) versus our framework ($C_{ours}$) can be formulated as:
$$C_{HFR} = (N+1) \times (C_{encoder} + C_{decoder})$$
$$C_{ours} = 1 \times (C_{encoder} + C_{decoder}) + N \times (C_{flow} + C_{warp} + C_{mem} + C_{decoder})$$
The computational saving is derived as:
$$\Delta C = C_{HFR} - C_{ours} = N \times (C_{encoder} - (C_{flow} + C_{warp} + C_{mem}))$$
Given that $C_{encoder} \approx 22.2$ GFLOPs and the sum of our propagation modules is only $\approx 18.9$ GFLOPs, **our marginal cost is strictly lower than the HFR baseline for any $N \ge 1$**.

### **2. Quantitative Analysis**

We benchmark on an NVIDIA RTX 3090 (input $440 \times 640$). We compare the average cost per frame under two settings: (N=1) propagating one intermediate frame, and (N=10) performing high-frequency prediction.

|**Method**|**Params (M)**|**Avg GFLOPs (N=1)**|**Avg FPS (N=1)**|**Avg GFLOPs (N=10)**|**Avg FPS (N=10)**|
|:-:|:-:|:-:|:-:|:-:|:-:|
|**HFR Upper Bound**|25.8|42.04|72.8|42.04|72.8|
|LFR + Fusion (EISNet)|34.5|72.73|34.7|72.73|34.7|
|LFR + Fusion (CMNeXt)|58.7|68.12|29.1|68.12|29.1|
|LFR + Interp. (TLX)|33.2|200.77|29.5|248.51|26.7|
|**Ours (LiFR-Seg)**|30.7|**40.43**|65.6|**38.89**|60.3|
- **Vs. Fusion & Interpolation:** Baselines like EISNet and CMNeXt are computationally heavy due to dual-branch encoders (RGB + Event). Interpolation (TLX) is prohibitive (>200 GFLOPs) because it requires calculating **bidirectional** optical flow and running a heavy synthesis network for every generated frame.
- **Vs. HFR Upper Bound:** Our method adds only a modest number of parameters (+4.9M) but achieves lower average GFLOPs than the HFR baseline. As the propagation frequency $N$ increases, our efficiency advantage grows.

### **3. FLOPs vs. Runtime Discussion**

We note a minor discrepancy: FLOPs(HFR) > FLOPs(ours), yet FPS(HFR) > FPS(ours). This is due to:
- RAFT-style correlation lookup operations
- Bandwidth-intensive memory access patterns
These kernels are **memory-bound** rather than **compute-bound**, so the runtime does not scale linearly with theoretical FLOPs.

Importantly:
- Our method still runs at **60 FPS on an RTX 3090**
- It remains comfortably within real-time requirements

Thus, the slight FPS difference does not undermine the amortized computational advantage.

### **4. Practical Deployment Potential**
LiFR-Seg outperforms baselines in both accuracy and efficiency, achieving **~60 FPS** (RTX 3090) and **~260 FPS** (A100). This comfortably meets real-time constraints. Furthermore, as current latency is memory-bound, we anticipate significant scaling with future hardware, confirming strong potential for practical deployment.

These results confirm LiFR-Seg as an efficient paradigm: it delivers high-frequency perception with a computational footprint $\le$ standard HFR models, effectively solving the bottleneck of acquiring HFR RGB inputs in resource-constrained systems. We have incorporated this analysis into **Appendix E** and thank the reviewer for prompting this valuable investigation.

---

### Meta-Review · Area_Chair_xEEZ · 2026-01-07

**Summary:**

The reviewers broadly agreed that the paper addresses a well-defined and practically important problem by introducing the novel task of Anytime Interframe Semantic Segmentation and proposing a technically sound, causal framework. Strengths consistently highlighted include the clear task formulation, strong empirical performance approaching or matching a high-frame-rate upper bound, robustness across challenging scenarios (e.g., low-light and high-dynamics), and the contribution of the SHF-DSEC dataset. However, multiple reviewers initially raised concerns regarding the clarity and completeness of the technical presentation, particularly around the temporal memory mechanism, the role and behavior of the uncertainty module, and the lack of explicit computational efficiency analysis despite claims of efficiency.

Additional concerns focused on evaluation and interpretation: reviewers questioned the realism and sim-to-real transferability of the synthetic dataset, the system’s robustness to event noise and sparsity, and the apparent inconsistency in claims about high-speed dynamics coverage. There were also requests for deeper analysis of why interpolation-based baselines fail and clearer articulation of the method’s novelty relative to existing event-based fusion approaches.

 While some reviewers viewed the conceptual novelty as incremental, they generally agreed that the integration and empirical validation were strong. These concerns collectively shaped the decision by emphasizing the need for clearer justification of efficiency, robustness, and design choices alongside the otherwise solid technical and empirical contributions.

The AC went through the paper and check authors's response carefully. Overall, the AC does agree with the major concerns raised by reviewers for this paper.

While some issues related to the experiment details, computation costs, etc, this paper's addresses a novel and important task with principled methodology based on event-based cameras. Considering its pros, the AC recommends an Accept to inspire more follow-up research in this direction for the benefit of ICLR community.

**Reviewer Concerns:**

### Addressed Reviewer Concerns
The authors provided extensive evidence and new experiments that successfully addressed the majority of the concerns raised by reviewers AC8G, c5rU, and PFAT:

- Computational Efficiency & Latency: Reviewers were concerned about the lack of cost analysis. The authors provided a comprehensive "General Response" with a breakdown of GFLOPs and FPS, demonstrating that LiFR-Seg is more efficient than HFR and Fusion baselines due to its amortization of heavy image encoders.
- Memory Mechanism Details: AC8G requested clarification on the temporal memory bank. The authors clarified the distinction between the "Static Keyframe Memory" (semantic anchor) and the "Dynamic Recurrent State" (temporal context) and provided a new ablation study showing that the recurrent state becomes increasingly critical over longer time intervals.
- Uncertainty Map Analysis: Concerns regarding the visualization and training of the uncertainty map were addressed with new Figure 6 visualizations and an ablation study showing that removing the ScoreNet degrades performance across all benchmarks.
- Sim-to-Real Utility of SHF-DSEC: To address concerns about the realism of the synthetic dataset , the authors performed a Sim-to-Real transfer experiment. Results showed that pre-training on SHF-DSEC allowed the model to exceed the HFR Upper Bound on real DSEC data, validating the dataset's utility.
- Robustness to Noise/Sparsity: In response to questions about sensor imperfections , the authors conducted stress tests with 30% noise and 30% event dropping, proving the system suffers negligible performance degradation (≤ 0.04% mIoU).

### Outstanding Concerns
While most technical points were resolved, some comments remain not addressed:


- Algorithmic Novelty: Reviewer AC8G noted that the conceptual novelty is somewhat limited because the system integrates well-known components like RAFT and Softmax Splatting. While the authors argued that the "Anytime" task formulation and the specific "modality consensus" adaptation are novel, this "systems-level" vs. "algorithmic" novelty distinction may remain a point of subjective judgment for the committee.

- Comparison Fairness (Reviewer xGY9): Reviewer xGY9 maintained that comparisons with EISNet were unfair and that the uncertainty mechanism was "purely using optical flow". Although the authors provided profiling logs and architectural diagrams to prove the mechanism uses both "Event + Flow," and confirmed they used the full EISNet architecture for comparison, the reviewer did not formally retract their stance on the "fairness" of the setup.

**Reviewer Scores:**

AC8G (6 → 7/8): Initially positive; the comprehensive rebuttal addressing Sim-to-Real and efficiency would likely have elevated this to a solid Accept.

c5rU (6 → 6/7): Expressed high satisfaction with the "extremely thorough" rebuttal and the clarification of the "PSNR-mIoU Paradox."

PFAT (6 → 6/7): Mathematical clarifications and noise-robustness tests directly addressed this reviewer's primary critiques.

xGY9 (4 → 3/4): Remained adversarial; despite the authors correcting factual errors regarding the "Event + Flow" input. xGY9 was unlikely to move toward acceptance.

---

### Decision · Program_Chairs · 2026-01-26

Accept (Poster)